# Study protocol for statin web-based investigation of side effects (StatinWISE): a series of randomised controlled N-of-1 trials comparing atorvastatin and placebo in UK primary care

Emily Herrett,[1] Elizabeth Williamson,[1] Danielle Beaumont,[1] Danielle Prowse,[1] Nabila Youssouf,[1] Kieran Brack,[1] Jane Armitage,[2] Ben Goldacre,[3] Thomas MacDonald,[4] Tjeerd van Staa,[5,6] Ian Roberts,[1] Haleema Shakur-Still,[1] Liam Smeeth[1]

For numbered affiliations see end of article.

**Correspondence to**
Professor Liam Smeeth;
liam.smeeth@lshtm.ac.uk

## ABSTRACT

**Introduction** Statins are effective at preventing cardiovascular disease, widely prescribed and their use is growing. Uncertainty persists about whether they cause symptomatic muscle adverse effects, such as pain and weakness, in the absence of statin myopathy. Discrepancies between data from observational studies, which suggest statins are associated with excess muscle symptoms, and from randomised trials, which suggest no such excess, have caused confusion. N-of-1 trials offer the opportunity to establish whether muscle symptoms during statin use are caused by statins in particular individuals.

**Methods and analysis** This series of 200 randomised, double-blinded N-of-1 trials in primary care will determine (1) the effect of statins on all muscle symptoms and (2) the effect of statins on muscle pain that is perceived to be statin related. Patients who are considering discontinuing statin use due to muscle symptoms and those who have discontinued in the last 3 years due to such symptoms will be recruited. Participants will be randomised to a sequence of six 2-month treatment periods during which they will receive atorvastatin 20 mg per day or matched placebo. On each of the last 7 days of each treatment period, participants will rate their muscle symptoms on a Visual Analogue Scale (VAS). At the end of their trial, participants will be shown numerical and graphical summaries of their own symptom data during statin and placebo periods. The primary analysis on the aggregate data from all participants will be a linear mixed model for VAS muscle symptom score, comparing scores during treatment with statin and placebo.

**Ethics and dissemination** This trial received a favourable opinion from South Central-Hampshire A Research Ethics Committee. Results will be published in a peer-reviewed medical journal. Dissemination of results to patients will take place via the media, website (statinwise.lshtm.ac.uk) and patient organisations.

**Trial registration number** ISRCTN30952488.

## Strengths and limitations of this study

► This trial will determine whether statins cause muscle symptoms during statin use in these participants, allowing clinician and participant to make an informed decision about continued use.
► This trial will evaluate a novel pathway of care for general practitioners to determine the cause of muscle symptoms during statin use.
► Patients who have previously experienced intolerable muscle pain during statin use may be unwilling to participate in the trial, therefore results cannot be generalised to them.
► The study will only assess the effects of a single statin at a single dose.
► The study will only capture symptoms arising within the 2-month treatment period.

## INTRODUCTION

Statins reduce cardiovascular disease (CVD) risk[1] and are widely recommended as part of the strategy for primary and secondary prevention of CVD.[2–4] Although statins are widely prescribed,[5] there is uncertainty about adverse effects.[6 7] Severe adverse effects (rhabdomyolysis, myopathy, diabetes mellitus and haemorrhagic stroke) are rare[8 9]; however, there has been widespread reporting in the lay and scientific media of other less well-defined statin-related symptoms, particularly muscle pain in the absence of myopathy (ie, symptoms with raised creatine kinase (CK) blood levels). These reports have been prompted by data from non-randomised, non-blinded observational studies,[10–12] but randomised controlled trials (RCTs) have

BMJ

found no evidence of an effect on these symptoms.[13] A recent review of the evidence from randomised trials and observational studies suggested that symptomatic adverse events may be misattributed to statins,[9] and there is further evidence from trials of statins of this misattribution.[14]

Uncertainty about the association between muscle symptoms and statins persists due to limitations of observational studies and trials. A major limitation of observational studies is a lack of blinding; patients taking a medication expect to experience adverse effects,[15] and therefore reporting of symptoms in statin users may be higher than in a comparable population not on statins. This phenomenon, the 'nocebo' effect, can lead to bias in unblinded studies.

In trials, outcome definitions have been inconsistent,[6] and the temporal nature of muscle symptoms related to statins may not be compatible with the nature of adverse event collection. Another important consideration is dilution bias; even in the absence of statins, musculoskeletal symptoms are very common among the age groups of people most likely to take statins and could dilute any true effect of statins on muscle pain. In the randomised Heart Protection Study, around one-third of participants in the placebo arm reported unexplained muscle pain or weakness in response to regular direct questioning from the study nurse, with an almost identical proportion among participants in the statin arm.[16] Dilution of statin-related symptoms by non-statin-related symptoms would have led to a bias towards the null.[17] Overcoming this bias and disentangling statin-related from non-statin-related muscle symptoms is key in determining whether statins cause muscle symptoms.

Despite evidence-based recommendations about the risks and benefits of statin use, many patients believe that their muscle symptoms are due to statins and discontinue use, therefore potentially missing out on the potential benefits. Clinicians faced with patients presenting with muscle symptoms during statin use are encouraged to measure the blood CK, which is substantially elevated in the rare cases of statin-associated myopathy, but in the vast majority of patients, the CK level will be normal, ruling out myopathy. In these cases, there is currently no other diagnostic tool allowing clinicians to empirically evaluate whether symptoms reported by an individual statin user are caused by the statin itself or by the 'nocebo' effect.

Therefore, this series of N-of-1 trials aims to offer the opportunity for individual patients to discover whether the symptoms that they are experiencing are attributable to statins. Each patient acts as their own control, and the treatment that minimises their symptoms can be established.[18 19] The proposed trial will address some of the criticisms of previous evidence. The trial is double-blinded and placebo-controlled to minimise bias, and the sequence of statin and placebo treatments is randomised. Additionally, the within-patient comparisons of symptoms experienced while on placebo and statin will allow us to determine (1) the effect of statins on all muscle symptoms and (2) the effect of statins on muscle pain that is perceived to be statin related.

## METHODS

### Study design

A series of randomised, double-blind, placebo-controlled N-of-1[18 19] trials in primary care.

Each participant in the study can be seen as having their own RCT, and StatinWISE is a series of 200 of these, which is equivalent to a single multiple-crossover study of 200 patients.

N-of-1 trials can be demanding on the time of both patient and clinician, and StatinWISE has been designed with both of these groups in mind. As described in the methods that follow, general practitioners (GPs) are asked to perform as few trial-related tasks as possible and are asked to follow standard care in the follow-up of participants. Trial visits and procedures for participants themselves are minimised, and outcomes are collected using a choice of methods.

### Study population

This study is taking place in primary care across England and Wales. Participating GP practices will recruit eligible patients from two groups as follows:
1. patients who are considering discontinuation of their statin due to muscle symptoms,
2. patients who have stopped taking statin in the last 3 years due to muscle symptoms.

### Inclusion criteria
- Adults (aged 16 years and above).
- Registered in a participating GP practice.
- Previously prescribed statin treatment in the last 3 years.
- Stopped or is considering stopping statin treatment due to muscle symptoms.
- Provided fully informed consent.

### Exclusion criteria
- Patient has any previously documented serum alanine aminotransferase (ALT) levels at or above three times the upper limit of normal.
- Patient has persistent, generalised, unexplained muscle pain (whether associated or not with statin use) and blood CK levels greater than five times the upper limit of normal.
- Patient has any contraindications listed in the Summary of Product Characteristics (SmPC) for atorvastatin 20 mg, including pregnancy.
- Patient should not participate in the trial in the opinion of the GP.

Patients who have not had a CK and ALT test within the previous 3 months will be required to undergo these tests prior to randomisation to ensure eligibility.

### Recruitment

Participants will be recruited directly from GP Practices or by advertising to the public.

(1) *Patients who are considering discontinuation of their statin due to muscle symptoms:* These patients will be invited to take part in the trial when they visit the GP to report muscle symptoms believed to be associated with statins, and where the patient/GP is considering stopping statins because of the muscle symptoms. The GP or research nurse will approach the patient with an invitation to take part in the trial.

(2) *Patients who have stopped taking statin in the last 3 years due to muscle symptoms:* A search of the practice electronic records will be performed on a two monthly basis for 1 year (or until recruitment targets are reached) to identify potentially eligible patients. The list will be reviewed by the GP to confirm clinical eligibility before patients are invited to take part.

A letter inviting them to attend a screening visit, accompanied with the patient information sheet (online supplementary appendix 1) for the patient to consider, will be sent by the trial team from their GP practice.

(3) *Patients who contact the CTU from advertising:* Patients who contact the CTU in response to advertising material will be sent a letter to request their GP details on a reply slip. Following receipt of these documents the CTU will contact their GP with their consent. The GP will be asked to confirm that the patient is potentially suitable for the trial and to provide brief clinical information to allow eligibility to be assessed. This information will then be provided to the GP surgery responsible for recruiting the patient.

Patients will have the opportunity to ask the research nurse any questions during the screening visit. The nurse will ensure that the patient understands the study and will record their informed consent (online supplementary appendix 2).

### Assignment of interventions

Consenting patients eligible for inclusion will be randomised by the research nurse/GP practice trial team using the online London School of Hygiene and Tropical Medicine (LSHTM) clinical trials unit (CTU) randomisation system, which will allocate them to a sequence of blinded placebo and atorvastatin treatment periods. There are six treatment periods each of 2 months' duration (each treatment period is exactly 8 weeks in duration). Each individual will be randomised to three paired blocks of treatment (either statin then placebo or placebo then statin), which is equivalent to randomisation, with equal probability, to one of the eight sequences shown in table 1 (with p=placebo and S=statin). There is no washout period at study initiation.

Randomisation codes will be generated and secured by the Information Technology team at LSHTM CTU, which has procedures to ensure that the trial team remains blinded. The codes will be made available to a Good Manufacturing Practice (GMP) certified clinical trial supply company explicitly for the treatment packs to be created in accordance with the randomisation list.

**Table 1** Treatment sequences in StatinWISE (S=statin and p=placebo)

| | Treatment period | | | | | |
| --- | --- | --- | --- | --- | --- | --- |
| | 1 | 2 | 3 | 4 | 5 | 6 |
| Sequence 1 | S | P | S | P | S | P |
| Sequence 2 | S | P | S | P | P | S |
| Sequence 3 | S | P | P | S | S | P |
| Sequence 4 | S | P | P | S | P | S |
| Sequence 5 | P | S | S | P | S | P |
| Sequence 6 | P | S | S | P | P | S |
| Sequence 7 | P | S | P | S | S | P |
| Sequence 8 | P | S | P | S | P | S |

### Blinding

The participant, general practice staff and trial team will all be blind to the participant's sequence allocation. Placebo will be manufactured specially to match the atorvastatin by a GMP certified manufacturer. Capsules and packaging will be identical in appearance for both active treatment and placebo. Participants will be asked to swallow the capsule whole without chewing or breaking it to minimise the risk of unblinding. The blinding process and first stage Qualified Person release will be done by the designated clinical trial supply company.

Unblinding will be available if a clinician believes that clinical management depends importantly on knowledge of whether the patient is currently receiving statin or placebo. In these cases, a 24-hour telephone service will be provided with the unblinding notified directly to the clinical requesting it.

### Interventions

The trial treatment consists of once per day oral administration of atorvastatin (20 mg) capsules or a matching placebo capsule (microcrystalline cellulose). The treatment phase will be 1 year for each participant. Participants will receive a 2-month (8 weeks) supply of trial treatment by post, through six treatment periods, and asked to take one capsule per day, swallowed whole. Two months is sufficient time for symptoms to appear in most patients[20 21] and to washout from the previous treatment period (median time to symptom improvement was 2 weeks following cessation).[22]

### Modifications to the trial treatment

For participants who experience intolerable muscle symptoms, the GP will be asked to follow standard care guidelines and measure the participant's CK and alanine transaminase. If the CK and ALT are within the normal range and the participant remains eligible, they can be offered the following options by the GP:

1. Continue with their current trial treatment.
2. Reduce the frequency to every other day.
3. Stop for that treatment period and resume at the start of the next period.

A participant is free to change their mind about participation at any time and would be advised to see their GP to discuss future routine care. A GP can withdraw a participant at any time if concerns arise or if the participant presents with any reason to stop atorvastatin as described in the SmPC for atorvastatin 20 mg.

### Monitoring adherence to the trial treatment
Adherence to trial treatment will be assessed by: (1) self-reporting, as part of outcome data collection and (2) counting pills remaining in returned packages (participants will be asked to return any empty or unused pill packets using stamped addressed envelopes provided with their treatment packs).

### Concomitant care
Throughout the trial, continued participant care will be at the discretion of their GP. In primary care, the participant will be recorded as having an ongoing statin prescription.
▶ Where treatment with an interacting drug is needed that will be for less than 1 month duration, the participant will be asked to stop the trial treatment for that period.
▶ Where treatment with an interacting drug is needed for longer than 1 month, the participant will be asked to withdraw from trial treatment completely.

Participants will be provided with an alert card that identifies them as a being randomised in StatinWISE. Participants will be asked to present this card to anyone providing medical care outside of their usual GP practice. The card will have a link to the trial website and their GP practice contact number.

### Optional genetic study
Participants will also be invited to contribute to a larger study that aims to identify genetic variants associated with adverse effects of statins, for which 9 mL of blood is required. Participation is optional and will not affect involvement in the main trial.

## OUTCOMES
### Primary outcome
The primary outcome is self-reported 'muscle symptoms', defined as pain, weakness, tenderness, stiffness or cramp to the body of any intensity; these are the symptoms most commonly reported by patients and are often the reasons for discontinuation. Though this primary outcome has a broad definition, making it potentially vulnerable to dilution bias ([17] and as discussed above), it has the key advantage of allowing participants to report any muscle symptoms, without constraining them to report only those that they are confident are statin related. Measures to deal with dilution bias are discussed in the Secondary analyses section.

The primary outcome will be assessed by the mean difference in Visual Analogue Scale (VAS) scores between treatment periods with atorvastatin and treatment periods with placebo, estimated via a linear mixed model.

In the seventh week of each 2-month treatment period, participants will receive reminders to alert them that follow-up data collection is approaching. Symptom scores on the VAS will be collected daily in the eighth week of each treatment period. Participants can choose to receive daily reminders on each day their data is due to be collected. Non-responders will automatically receive a reminder from the trial team after 24 hours of the due date.

### Secondary outcomes
Secondary outcomes (box) relate to participant belief about the cause of their muscle symptoms, the site of muscle symptoms, how the muscle symptoms affect the participant and information about any other symptoms.

A key secondary outcome is pain for which the participant answers 'yes' or 'don't know' to the question 'Do you think these muscle symptoms are related to your study medication?', asked at the end of each treatment period. People answering 'no' to this question would have their VAS scores set to zero for that treatment period for this specific secondary outcome.

Secondary outcome number three in box were taken from the Brief Pain Inventory.[23] Other secondary outcomes are adherence to medication, the participant's decision about statin treatment following the trial

---

**Box   Secondary outcomes**

1. Whether or not participants with muscle symptoms during each 2-month period believe that their symptoms were caused by the study medication, comparing periods of statin treatment with placebo.
2. Site of muscle symptoms (single or multiple; head and neck/upper limbs/lower limbs/trunk).
3. Among participants reporting muscle symptoms, the Visual Analogue Scale scores (range 0–10) for the following, comparing periods of statin treatment with placebo:
   a. General activity
   b. Mood
   c. Walking ability
   d. Normal work (includes both work outside the home and housework)
   e. Relations with other people
   f. Sleep
   g. Enjoyment of life
8. Other symptoms that the participant believes can be attributed to the study medication (grouped: musculoskeletal, gastrointestinal, respiratory, neurological, psychological and other).
9. Adherence to study medication as assessed by: (1) self-report and (2) counting pills remaining in returned packages, and the relationship between adherence and muscle symptoms.
10. Participant decision regarding future statin use and the relationship to their primary outcome.
11. Whether participants found their own trial result helpful in making the decision about future statin use.

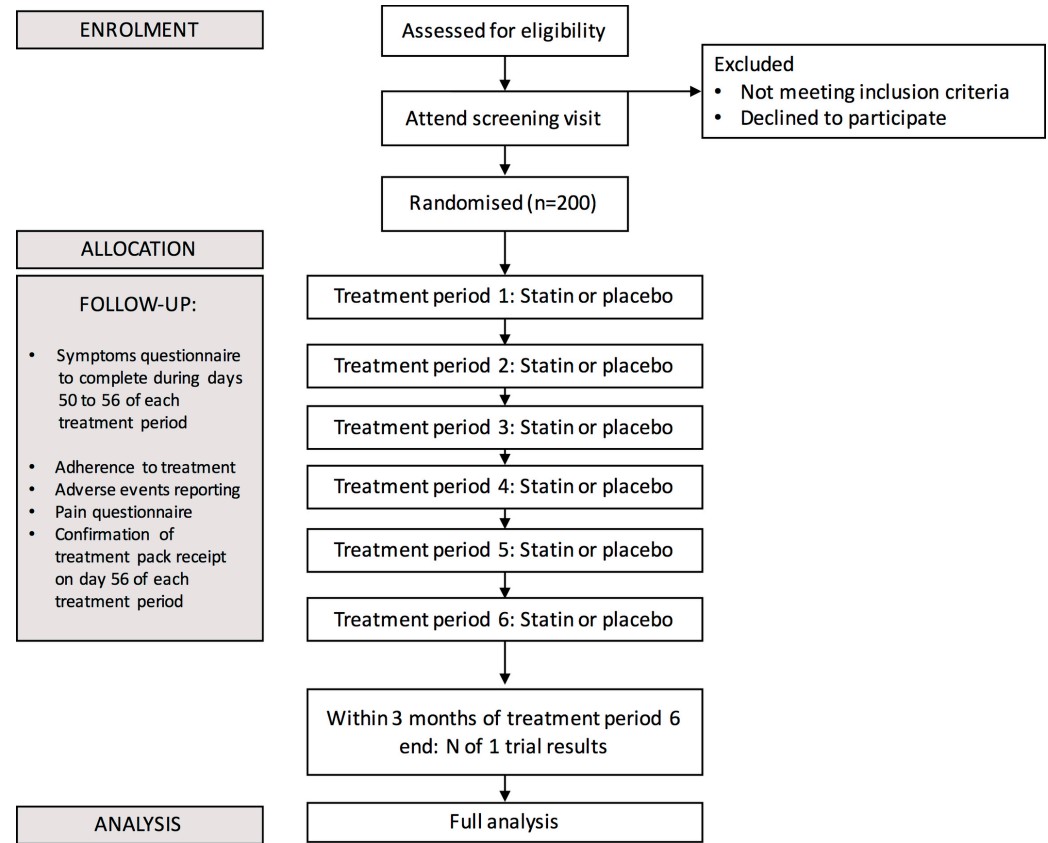

**Figure 1** Trial overview.

### Data collection methods
#### Baseline
Baseline data will include demographic information, contact details, an eligibility assessment and general medical history. These data will be collected at each GP practice by the GP or research nurse and will be entered directly into the online bespoke trial database provided by the LSHTM CTU.

Participants will choose the method of data collection most suitable for them, from the following:
1. Bespoke mobile app, which will require participants to use their own smartphone.
2. Online database using a computer, phone or tablet.
3. Paper forms which they will receive by post at the same time as their trial treatment and which they can complete themselves or they can request a trial team member to contact them by phone to help with completing the forms.
4. Trial staff will telephone the participant on each data collection day and complete the questionnaire based on the participant answers.

#### Treatment phase follow-up
Follow-up symptoms and adherence data will be collected directly from each participant using their preferred data collection method. Reminders will be sent to participants,

as described above, to encourage complete submission of data.

#### End of trial data
Participants, together with their GP, will receive their individual results within two months of the last treatment period and will discuss these with the GP or research nurse. Three months after the last treatment period (month 15), trial staff will email/telephone the participant to document their decision on future statin use and whether their results helped reach this decision.

### Participant timeline
See figure 1 and online supplementary appendix 3.

### Sample size calculation
The power calculations are based on being able to detect a 1 cm difference in the VAS pain score. This was chosen to represent the smallest VAS change in pain which patients would perceive as being beneficial and might therefore change a patient's decision regarding subsequent statin use. Two studies have concluded that the smallest change in VAS pain score corresponding to 'a little more' or 'a little less' pain was 1.3 cm, with a lower limit of the CI at 1 cm.[24 25] A 1.3 cm minimum change value was used to power a pilot series of N-of-1 trials for statin adverse effects,[26] and our study is therefore powered for a 1 cm change as a conservative estimate of the smallest beneficial change.

Using simulation, we estimated that a sample size of 64 participants provides approximately 90% power to detect a treatment effect of at least 1 cm, assuming a type I error of 5%. Allowing for loss to follow-up of 40% of participants through the trial inflates the required sample size to 107 participants.

Period effects (changes in underlying VAS pain score due to factors other than randomised treatment, for example, seasonal and activity-related), variability in individual statin effects across participants and imperfect adherence to the assigned treatment were investigated by further detailed simulations. VAS pain scores are not normally distributed, since they are bounded (0–10 cm) and can display large fluctuations in response. Therefore, further power calculations were performed drawing the outcome from a Beta distribution and from a distribution with normal variance components on a logit scale, to assess the robustness of the sample size estimates to the distribution chosen. These factors all have the effect of decreasing power, thus increasing the sample size required. An approximately 80% increase in the sample size required in the absence of these effects provided approximately 90% or more power across a plausible range of these potential effects, thus we determined that a final sample size of 200 was required.

### Statistical methods
#### Primary analysis
To estimate the population average estimate of the trial VAS muscle symptom score, data from each N-of-1 trial will be aggregated. We will adopt an intention-to-treat approach. Participants who enter data on muscle symptoms at least once during a treatment period with statin and at least once during a treatment period with placebo will be included in the primary analysis.

The primary analysis will be a linear mixed model for VAS muscle symptom score with random effects for participant and allowing the treatment effect to vary randomly across participants. Residual errors will be modelled using a first-order autoregressive error structure within each treatment period to account for correlation between the seven daily measurements, with robust standard errors to account for non-normality of the VAS scores. Although VAS muscle symptom scores are unlikely to be exactly normally distributed, analysing such data using normal-based methods is likely to be a sufficiently robust approach.[27] Sensitivity analyses will explore the robustness of conclusions to period effects, within-participant correlation structures and missing data. All tests will be two-sided. $P < 0.05$ will be considered statistically significant.

#### Secondary analyses
The secondary outcomes include a single binary measure of whether the participant reports having muscle symptoms during that treatment period or not. This will also be combined with the follow-up question pertaining to attribution, to obtain a single binary measure of whether the participant reports having muscle symptoms that they attribute to the trial treatment or not. These two binary outcome measures will be assessed using a logistic mixed model with random participant and treatment effects and fixed period effects.

We will also perform a secondary analysis using the daily VAS scores, in combination with the follow-up question pertaining to attribution. This analysis will be performed in the same way as the primary analysis, but with VAS scores set to zero if the patient attributes their symptoms to a non-statin-related cause.

By removing symptoms with a known cause, these secondary analyses will be restricted to symptoms that the participant believes could be due to their study medication. This could reduce possible dilution bias that may be present in the primary outcome. If the secondary analyses of muscle symptoms that the participant attributes to the study medication produce a substantially higher measure of effect than the primary analysis of all muscle symptoms, then this suggests that the primary result may have been affected by dilution bias.

Secondary outcomes relating to the impact of the atorvastatin treatment on other aspects of life will be analysed in a similar manner to the primary outcome, omitting the autoregressive correlation structure since these secondary outcomes are measured once per treatment period.

We will investigate whether the excess muscle symptoms, if any, experienced during treatment periods with statin are at a single body site versus multiple sites, and whether symptoms are more common at any specific body site.

Descriptive statistics will be used to summarise the measures of adherence to randomised treatment and their relationship to the statin and placebo periods. We will use the measures of adherence to randomised treatment to perform an efficacy analysis based around an instrumental variables approach.[28] Since these analyses require much stronger assumptions than the intention-to-treat analysis above, the results of the efficacy analysis will be presented and interpreted as a secondary analysis.

We will relate the participant's decision regarding future statin use, and whether or not the participant found their own result helpful in making their subsequent treatment decisions, to their individual estimated effect of the atorvastatin treatment.

#### Subgroup analyses
There are no a priori subgroup analyses planned. If an overall population-level effect is detected, we may investigate whether the effect varies within subgroups defined by measured baseline characteristics. These analyses will be presented and interpreted as being exploratory.

#### Data monitoring
An independent data monitoring committee (DMC) has been appointed for this trial to oversee the safety monitoring. The DMC will review accumulating data on a regular basis from the ongoing trial and advise the trial

steering committee (TSC) regarding the continuing safety of current participants and those yet to be recruited, as well as reviewing the validity and scientific merit of the trial. An independent statistician is appointed to provide the analysis service required by the DMC.

The intervention (atorvastatin) has a marketing authorisation in the UK and has been in clinical use for decades. Atorvastatin is not a new drug and has a well-documented safety profile. Furthermore, this trial is being conducted in a population in which statin use is clinically indicated. It is anticipated that participants in this trial are at higher risk of CVDs because of underlying clinical conditions requiring statin use. Additionally, as participation is for 15 months, it is likely that they will have common medical problems (eg, colds, coughs and fevers). On the basis that: (1) participants would have had prior exposure to the trial treatment, (2) the trial treatment is clinically indicated for their medical condition and (3) the known safety profile of the trial treatment, we will limit adverse events reporting to any untoward medical occurrence not listed in the Investigational Medicinal Product Dossier and/or SmPC affecting a trial participant which at any dose:

► results in death
► is life-threatening
► requires inpatient hospitalisation or prolongation of existing hospitalisation
► results in persistent or significant disability/incapacity
► is a congenital anomaly/birth defect.

These events will be reviewed routinely by the trial Medical Advisor and will be reported routinely to the DMC. There are no extra tests or procedures unless participants agree to contribute a DNA sample to the optional genetic study, for which a 9 mL blood sample is required.

We will use central monitoring[29] along with investigators' training and meetings and extensive written guidance to make sure the trial is carried out properly. We plan to carry out onsite monitoring where central statistical monitoring shows abnormality. Consent forms will be monitored centrally at the CTU.

Investigators/GP practices are required to provide direct access to source data/documents for trial-related monitoring, audits, ethics committee review and regulatory inspection. The majority of source data will be electronic from the web database and mobile app. All trial-related and source documents must be kept for 5 years after the end of the trial.

## Harms

Known serious adverse effects of statins (rhabdomyolysis and myopathy) are extremely rare. Patients who have previously experienced rhabdomyolysis or myopathy will be excluded from the trial, based on information provided from each participant's GP. This trial aims to address uncertainty about less severe side effects of statins, but participants experiencing intolerable symptoms should report these to their GP and a decision about whether to continue with the study can then be made.

## Auditing

The study may be subject to audit by the LSHTM under their remit as sponsor and other regulatory bodies to ensure adherence to GCP.

## Patient and public involvement

Three patient representatives are on the TSC, and a StatinWISE patient involvement group has provided feedback on the trial design, patient information sheet and data collection tools. The group will continue to be involved through the course of the trial and will play an important role in designing materials for dissemination.

## ETHICS AND DISSEMINATION
### Ethical issues
#### Approvals

Protocol amendments will be submitted for approval to the ethics committee, will be updated on trial registries and will be circulated to all study sites.

#### Informed consent

Informed consent will be obtained by the GP or research nurse (online supplementary appendix 2). Informed consent will also be requested for a blood sample for an optional genetic study and for that blood sample to be retained for use in future ethically approved studies.

#### Confidentiality

Participant data will be accessed only by authorised personnel from the LSHTM. All will have a duty of confidentiality, and no data will be disclosed outside the research site. All participant and GP practice-level information will be confidential.

#### Post-trial care

The follow-up period ends 15 months after the first treatment day with a final contact (telephone or face-to-face) with the research nurse. This will be considered as the end of trial for participants, and further routine clinical care will be provided by their GP.

### Dissemination plans

The trial results will be published in peer-reviewed journals. All publications will follow the Consolidated Standards of Reporting Trials statement.[30] Links to the publication will be provided in all applicable trial registers. Dissemination of results to patients will take place via the media, trial website (statinwise.lshtm.ac.uk) and relevant patient organisations. The results of the trial will be reported first to trial collaborators. Collaborating investigators will play a vital role in disseminating the results to colleagues and patients. Authorship for all publications will be based on the criteria defined by the International Committee of Medical Journal Editors.[31]

**Author affiliations**
[1]Department of Epidemiology and Population Health, London School of Hygiene and Tropical Medicine, London, UK
[2]Clinical Trial Service Unit, Nuffield Department of Population Health, University of Oxford, Oxford, UK
[3]Department of Primary Care Health Sciences, Centre for Evidence-Based Medicine, Radcliffe Observatory Quarter, University of Oxford, Oxford, UK
[4]Medicines Monitoring Unit and Hypertension Research Centre, University of Dundee, Dundee, UK
[5]Health eResearch Centre, Farr Institute, University of Manchester, Manchester, UK
[6]Division of Pharmacoepidemiology and Clinical Pharmacology, Department of Pharmaceutical Sciences, Faculty of Sciences, Utrecht Institute for Pharmaceutical Sciences, Utrecht University, Utrecht, The Netherlands

**Acknowledgements** We are grateful for the contributions and support of the following StatinWISE team members: trial coordinating team: Collette Barrow, Sergey Kostrov, Hakim Miah, Dr Lori Miller and Andrew Thayne; steering committee: Professor Michael Moore (Chair), Rebecca Harmston, Maurice Hoffman, Brian MacKenna and David Symes; and data monitoring and ethics committee: Professor John Norrie and Professor Nicholas Mills.

**Contributors** The manuscript was prepared on behalf of the StatinWISE trial team: LS, JA, DB, BG, EH, KB, TMD, DP, IR, HS-S, TvS, EW and NY. LS and BG conceived the trial idea. All authors contributed to the development of the trial protocol. EH and EW drafted the manuscript, and all authors contributed to its revision and gave final approval. EW provided statistical expertise.

**Funding** The research is funded by the National Institute for Health Research – Health Technology Programme (reference 14/49/159). The study is sponsored by the London School of Hygiene and Tropical Medicine, Keppel Street, London WC1E 7HT. The study is registered at controlled-trials.com ISRCTN30952488, August 2016, ClinicalTrials.gov with reference NCT02781064 and EUDRACT 2016-000141-31.

**Disclaimer** The views expressed are those of the authors and not necessarily those of the NHS, the NIHR or the Department of Health.

**Competing interests** None declared.

**Ethics approval** The study protocol (V.3.0 28th June 2016) has received a favourable opinion from the South Central-Hampshire A Research Ethics Committee, reference 16/SC/0324.

**Provenance and peer review** Not commissioned; externally peer reviewed.

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
