## [Reviewer comments · BMJ Open]

ARTICLE DETAILS

TITLE (PROVISIONAL)	Study protocol for Statin Web-based Investigation of Side Effects (StatinWISE), a series of randomised controlled N of 1 trials comparing atorvastatin and placebo in UK primary care
AUTHORS	Herrett, Emily; Williamson, Elizabeth; Beaumont, Danielle; Prowse, Danielle; Youssouf, Nabila; Armitage, Jane; Goldacre, Ben; MacDonald, Thomas; van Staa, Tjeerd; Roberts, Ian; Shakur-Still, Haleema; Smeeth, Liam

VERSION 1 – REVIEW

REVIEWER	Gabriel Chodick Maccabi Institute for Research & Innovation, Tel-Aviv University, Israel None
REVIEW RETURNED	25-Apr-2017

GENERAL COMMENTS	This is a well-written study proposal of N-of-1 trial aimed to assess the relationship between muscle symptoms and statins as well as patients' perceptions regarding this relationship. N-of-1 trial is a most effective way to address this issue with minimal biases and therefore, this publication might be of interest for readers. The planned study seems to be well powered for VAS difference of at least 1cm, both by number of patients and also by the number of treatment periods. 1. The study design is heavily dependent on the assumption that statins-related muscle symptoms are relived shortly after statins are discontinued and re-experienced if therapy is reinitiated. This might be the case; but more text is required on this strong assumption.2. Adherence with therapy is a major challenge in this population; did the authors consider measuring LDL levels at different periods to validate drug consumption? To keep blinding, patients should be allowed to get their LDL levels while under therapy.3. It is unclear whether a washout period is planned at study initiation phase.4. Patient safety: pregnancy issue should be explicitly mentioned5. Feasibility: N-of-1 trials are complex and may burden on the caregivers and research team. More details are required on how this should be effectively managed.
--

REVIEWER	Johann Auer Department of Cardiology and Intensive Care St. Josef Hospital Braunau Austria none
REVIEW RETURNED	15-May-2017

GENERAL COMMENTS	This study will investigate 200 patients in a randomised, double blinded N-of-1 design in primary care and will determine (i) the effect of statins on all muscle symptoms, and (ii) the effect of statins on muscle pain that is perceived to be statin related. Patients who are considering discontinuing statin use due to muscle symptoms, and those who have discontinued in the last three years due to such symptoms, will be recruited. Participants will be randomised to a sequence of six two-month treatment periods during which they will receive atorvastatin 20mg daily or matched placebo. On each of the last seven days of each treatment period, participants will rate their muscle symptoms on a Visual Analogue Scale (VAS). This issue is timely and the scientific question is worth to be investigated. The authors use a novel web-based setting. Participants will choose the method of data collection most suitable for them. In general, the design is appropriate and seems to be scientifically accurate. I have some minor points that should be addressed: The authors state that this is a „...series of randomised controlled ... trials...“. This may suggest that this is not one but more trials. However, this is one trial that included 200 patients in a N-of-1 design. This should be clarified to avoid confusion. The authors focus on muscle symptoms. However, statins are associated with other skeletal-related symptoms beyond muscles. Assessment of such symptoms would give further insights into this issue. Blinding may be compromised by (unscheduled) LDL-C. measurements (driven by the patient) during the study. How do the authors handle this issue?
---

REVIEWER	Beatrice A. Golomb, MD, PhD University of California, San Diego United States
REVIEW RETURNED	18-May-2017

GENERAL COMMENTS	The authors cite desirability of N-of-1 trials of statins in those with perceived statin symptoms for the purpose of determining whether muscle symptoms short of rhabdomyolysis can be caused by statins. Strengths of the study include use of N-of-one randomized double blind design. Another strength is the inclusion of a question whether the patient believes the symptom to be statin-related.
--

Based on data in an RCT we conducted, ~40% on placebo thought they were on statin, ~60% on statin did so – if the difference is similar, a larger sample may be needed. In principle, a properly conducted study of this kind could yield important information. And a write-up of such a study could be rendered acceptable. However, in its current form the manuscript (and accompanying study design) have significant limitations. Attention to these is necessary for the paper to be made suitable for publication – and for the underlying study to be scientifically and ethically viable. (Members of the ethics review board were presumably not experts on this issue, and were guided by the presentation of evidence – including omissions therein.)

1. Characterization of evidence regarding muscle adverse effects: The presence of such multiple lines of evidence affirming the existence of statin muscle problems (short of rhabdomyolysis) should be fully acknowledged. Evidence in support of this has already included (among other findings): double-blind N-of-one dechallenge-rechallenge biopsy evidence¹; unblinded dechallenge-rechallenge from which potency/dose-response evidence naturally emerged – in patients highly unlikely to be aware of potency relationships²; emergent potency evidence in FDA AERS data that matches the findings from the aforementioned dechallenge rechallenge data, and documents elevated relative odds ratios of reporting for muscle problems relative to other drugs³. RCT comparisons of higher-vs-lower statin potency further support dose dependence in CK elevations⁴. RCT evidence in which a gene variant, particularly coupled with higher potency statins, was linked not only to “myopathy” (rhabdomyolysis) but to what was termed “incipient myopathy” which was really two-organ problems encompassing muscle and liver dysfunction⁵ – but, encompassing muscle. One of the participants on this was an author on that study, which should not have been omitted.

2. Evidence has already documented mechanisms associated with (non-rhabdomyolysis) muscle problems due to statins, including oxidative stress (via statins’ prooxidant antioxidant duality⁶), mitochondrial/ metabolic mechanisms/ fatty acid oxidation abnormalities^{1, 7-9}, and autoimmune necrotizing myopathy¹⁰⁻¹². Genetic predispositions of a character that supports statin induction of metabolic myopathies has also been reported⁸. Additionally, dose-dependent withdrawal of coenzyme Q10 with statins (reducing coQ10 in parallel with the cholesterol on which it is transported, since both are products of the mevalonate pathway) is expected to unmask mitochondrial pathology⁹, and thus, in a subset, muscle symptoms, just as supplementation with coenzyme Q10 can relatively bypass any of a range of mitochondrial abnormalities¹³. The existence of such evidence should be fully acknowledged.

3. Based on the above, clearly, muscle symptoms other than rhabdomyolysis do occur, due to statins, and in any given patient, muscle symptoms arising on statins, and perhaps particularly if attributed by the patient to statins, may be due to the statin. If this study shows low rates of recurrence (even if the dose and duration issues are fixed), it will not inform us about those patients, because we cannot know how the character of patients who participate relates to the character of patients who do not.

4. Analogous to the case of glucose, muscle effects of statins are linked to oxidative stress and thus bidirectional changes may produce neutral net effects on statin relative to placebo – even when high dose or multiple risk factors increase risk. This point has been made elsewhere^{14, 15}. Whether statin problems are or are not due to the statin in other patients, does not obviate the possibility that statins causally produced the problem in this patient. This should be acknowledged.

5. Copious evidence shows a powerful relationship of conflict of interest to study outcome and interpretation¹⁶⁻²¹. As former BMJ editor-in-chief Richard Smith noted, in relation to findings of a study, “It suggests that, far from conflict of interest being unimportant in the objective and pure world of science where method and the quality of data is everything, it is the main factor determining the result of studies”²². In this vein, it is of concern that articles are primarily cited that, unless they address other points, seek to repudiate a statin relationship to muscle symptoms. This concern is magnified by the prior involvement of several of the authors/ investigators here, in articles that dismiss the existence of causal statin effects on muscle. In this context it is imperative that the authors also demonstrate that they are familiar with, and can represent, the best case for the contrary view. Otherwise there is the appearance that the study may be designed and implemented by parties – who have previously sought to discredit statin muscle adverse effects – potentially for the purpose of further discrediting their existence, if such effects do occur. And with any honest reading of the literature, coupled with basic methodological acumen, they do occur. Therefore I suggest the purpose be recast as assessing how frequently instances can be validated, among persons amenable to rechallenging, with the present design. (Considerations below of dose and duration still require attention.)

6. It is also vital that the design of the study is up to snuff, that it have the ability to detect muscle problems if they arise. If not, such a study may be frankly damaging, by fostering a false narrative and exposing additional patients to risk and morbidity. Several modifications to the design are in order.

Design

a. Dose: The authors must acknowledge that dose-potency is an issue. And they must adjust the design accordingly. Evidence for dose/ potency dependence of statin adverse effects (previously reviewed⁹), renders the present design problematic. Persons with muscle symptoms on statins who had discontinued the statin, then were rechallenged by their physician (unblinded) with a statin for at least two months, reliably reexperienced the problem – if the rechallenge statin was of higher expected potency than the original statin². The sample was self-selected. It is possible that those who did not reexperience the problem may have been less likely to respond. However, that would not account for the dose relationship found. This is unlikely to reflect expectation/ nocebo effects, not only because according to the Nordic Cochrane center such effects – in a controlled setting, largely do not exist (placebo is generally not different from no treatment, when there is randomization to both²³); but because even if they did, most physicians are/ were not closely familiar with potency equivalencies, much less were patients.

The recurrence rate was still substantial with lower statin potency (over 70%), at least among those who submitted adverse event reports to the patient-targeted database.

However in the modern era there is greater use of high potency statins – and it cannot be presumed that persons will experience statin problems on atorvastatin 20mg, who previously experienced problems on 80 or 40mg, or on rosuvastatin 40mg (or 20mg). So for the stated purpose, to examine whether statin problems are real, rather than whether there is a statin dose which patients can tolerate, it is important that an adequate potency statin be employed. (Though as above, the purpose needs to be revised.)

{Of note, “equivalent potency” but different statin agents may be problematic. First, potency equivalencies are inexact. Second, persons may have genetic or other factors (renal, hepatic, grapefruit juice, concurrent interacting medications) that may fluctuate over time, or that may affect clearance of one but not another statin, producing a higher (or lower) effective potency of the index statin than would be reflected in the subsequent “equivalent” potency statin. Changes in nutritional status, organ function, or concurrent oxidant exposures could influence experience of the problem differentially with reexposure (less, or more), producing different results. Such factors are less likely to be decisive when a higher potency agent is used for rechallenge. (Higher potencies may pose risk of greater problems – of course, that is for those of us who acknowledge the existence of such problems.) The design must adequately consider this. It would clearly be unethical to conduct a study of this kind in a fashion that might disrespect patients’ contribution by subjecting them to risk, while missing a real problem.

Therefore, the study design must be revisited to better accommodate dose and duration considerations.

b. Duration: The two month time period may seem at first glance to be adequate but it is not, and particularly not with a multiple on-off design. Two months is consistent with the duration used by the Phillips 2002 double-blind N-of-1 placebo-controlled dechallenge-rechallenge biopsy study – which it is, incidentally, inexcusable not to cite, for a study of such related design¹. However it should be noted that Dr. Phillips later found that it was not every patient for whom a two months period sufficed, based on patients in his statin myopathy clinic (personal communication). The two month time might also appear to comport with the two month time period in our analysis of percent recurrence of statin muscle symptoms with rechallenge². But our recurrence analyses, we considered those who had rechallenged for at least two months, and most who had been back on statin for at least two months had, naturally, been on statin for more than two months at the time they completed the survey. Moreover, our direct data clearly show that two months is not consistently sufficient. Direct data on patient-reported time to onset and recovery were provided in that study. Muscle symptoms do not necessarily follow the timecourse of statin effects on lipids, analogous to the fashion in which symptoms caused by hypoxemia outlast the duration of the hypoxemia and may be permanent. Mean time to first noted improvement following statin discontinuation was 8 weeks, after first statin usage leading to problem – muscle adverse effect or MAE – with maximal recovery taking a mean of 29 weeks (though, median times were shorter, at 2 and 8 weeks respectively).

Confining analysis to those who provided timecourse data with first statin leading to MAE and statin rechallenge leading to MAE, mean time to recovery after first statin muscle AE was six weeks; but was twice as long – 12 weeks – following the rechallenge statin. Also, though the median time to recurrence with rechallenge was significantly shorter than with the first statin causing a MAE, some people nonetheless took months for recurrence (of muscle symptoms that also resolved with rechallenge, so were likely statin associated). If longer times are not permitted for onset and resolution, the study will ensure that symptoms, due to the statin will be difficult for patients to distinguish during the placebo and statin periods. I would advise that if the stated purpose is to assess whether problems like these are real, then data relevant to real-world timecourse be used to determine timecourse for the study, and to better assess study power based on the fraction expected to conform.

c. Participants: Since the stated purpose is to clarify whether statins can cause muscle problems, or if problems reported by patients are legitimate, and the investigators purports to show an avenue physicians can use, participants should be restricted to those who i) reported experiencing muscle problems on statins at least twice (with dechallenge-rechallenge), and ii) did so on a clearly lower potency statin without at the time being on an interacting medication; and iii) in a less than 1.5 month time period (since participants begin answering questions before two months). The study is not meaningful if participants are exposed to a potentially lower potency agent than the one that caused the problem (due to the agent or concurrent exposures); or took a longer time; and it is important to ascertain the degree of need for a placebo-controlled blinded study, which will be quite costly. The need may be negated if unblinded dechallenge-rechallenge was relatively reliable.

d. Secondary outcomes: None are chosen to reflect the more commonly reported adverse effects. Sleep comes closest. Cognition, fatigue, neuropathy, and irritability are among the most common problems reported by patients, in a patient targeted database. Three have RCT studies supporting them, and neuropathy has both good quality dechallenge-rechallenge evidence with concurrent objective findings (alluded to elsewhere); and a large Danish national database study²⁴. Even though your study focuses on muscle symptoms, data collected on other adverse effects should be based on past experience so that the study does not underreport the potential impact of rechallenging on a statin. (Sleep can be retained.) Since the sample size required to show problems with, say, sleep was previously on the order of 1000 participants, any one of these problems is unlikely to be significantly increased; but you should look at the count of these particular symptoms reported, on statin vs placebo.

e. Include narrative comments from patients about their experience. Allow them to comment on whether they feel the experience was more helpful or more harmful. (You only ask if it was helpful; for a study examining harm, it is critical to ask about harm.)

f. Since this is a study of persons who reported past muscle adverse effects, to see if they will report them again, CK assessment is de riguer, at baseline (before any treatment or placebo, ideally twice),

and at the conclusion of each treatment period. I would suggest cardiopulmonary function testing of the kind innovated by Dr. Phillips, particularly respiratory exchange ratio, as well²⁵.

Related to this, "Patients who have previously experienced rhabdomyolysis or myopathy will be excluded from the trial." Please clarify, how is myopathy and how is rhabdomyolysis defined for this purpose? How will you ascertain if these occur in patients?

g. Long term follow-up is imperative to include. I would also urge revision of the methods to accommodate a randomized pure placebo arm (or based on equivalence to this from Nordic Cochrane data, an arm randomized no-treatment, for the duration of the study), to assess whether subjecting patients who have had possible prior statin problems, to subsequent statin courses may accelerate subsequent physical decline.

Several forms of evidence support the possibility of persistent physiological impairment even if there is clinical resolution, following statin adverse effects. These include: persistent biopsy findings of myopathy in a sizeable subset²⁶; and persistent cardiopulmonary abnormalities^{25, 27}. (These latter presumably also reflect predisposition – but the existence of predisposition is contingent on the possibility that there was in fact something to predispose toward. Additionally, related changes in normal patients on statins goes against this serving purely as predisposition²⁸). Further factors pointing to persistence include shorter mean time to recurrence following rechallenge; and longer mean time to noting improvement following rechallenge's dechallenge (reviewed above); and partial persistence of symptoms in a sizeable subset (as above). Additionally, after statin neuropathy, which can arise with or independent of myopathy, there is "Clinical evidence of proximal and distal weakness and muscle fasciculations and persistent abnormalities of sensory conduction after recovery"²⁹.

h. Participants should, ethically, be informed that problems arising on statins do not consistently resolve fully, including muscle symptoms^{2, 8, 30, 31}. They should be advised that there have been a number of reports in which problems arising on statins resolved – clinically at least – after each (one or more) prior discontinuations, but then after the problem developed on the next statin, the problem has failed to recover, or recovery fully, on discontinuation. (This has also occurred for people who previously experienced no problem on a statin³¹.)

i. Ethics and Information to Participants:

i) Ethically, participants should be informed that there are no data to indicate the presence of net benefit of statins in persons who have experienced statin muscle problems. They should be advised that, even among those who have not had adverse effects (those who have had adverse effects may fare worse), the expected median survival benefit on statins may be quite low – according to one study, perhaps on the order of days, including for secondary prevention³². They should be informed of the range of reported adverse effects that can arise on statins.

ii) Participants should be informed that problems arising on statins do not consistently resolve fully, including for muscle symptoms^{2, 8, 30, 31}.

They should be advised that there have been a number of reports in which problems arising on statins resolved – clinically at least – after each (one or more) prior discontinuations, but then after the problem developed on the next statin, the problem has failed to recover, or recovery fully, on discontinuation³¹.

iii) It was stated that “Known serious adverse effects of statins (rhabdomyolysis and myopathy) are extremely rare.”

a) This is not a meaningful statement, because of effect modification. The rarity of problems depends entirely on characteristics of the patient (and treatment). Claims data, for instance, showed that the number needed to treat to have one case of rhabdomyolysis ranged from one in 22,727, to one in 484 for older patients with diabetes mellitus on statin plus fibrate, to one in 9.7 to 12.7 treated patients in a special treatment setting³³ – that setting involving the highest frequency will not occur now, but there will be other specialized subsets that will have markedly elevated risk for serious problems due to a confluence of risk factors, including some we may not have identified. {Similarly our RCT data affirmed effect modification – again showing that subsets – again, older age coupled with metabolic syndrome factors – had markedly elevated risk of glucose elevation due to statin (i.e. compared to placebo); and while older age plus female sex (each a known risk factor for statin problems) markedly increased risk of muscle weakness – replicated on each simvastatin vs placebo and pravastatin vs placebo, so not plausibly attributable to a chance finding due to multiple looks.

b) Second, it is vital to be clear to patients that by the definition employed, ongoing suffering from pain, for instance, is not counted as “serious.”

j. Exclusions include “Patient has persistent, generalised, unexplained muscle pain (whether associated or not with statin use) and blood creatine kinase (CK) levels greater than 5 times the upper limit of normal”

a) As elsewhere in this review, patients may have serious statin-induced myopathy without CK elevations.

b) Additionally, some prior “muscle” problems may be due to neuropathy^{24, 29, 34}, not associated with CK elevation. Risk of these problems may particularly accrue with time of statin use^{35, 36}, and though many or most recover partially³⁷, many do not recover fully³⁸; moreover, even where there is clinical recovery, there may be persistent pathological injury²⁹; and one prospective study found development of pathological changes, even in absence of clinical ones³⁹.

c) These patients – at potentially heightened risk of statin adverse effects (based on prior possible adverse effect) – should be provided with this information prior to accepting rechallenge on statins.

Analyses:

Sample size calculations are problematic. The study is powered based on the notion that a 1cm VAS difference is relevant, and that that must be the average effect.

But the whole premise of the study is that not everyone in whom muscle problems developed on statins, developed these due to statins, ANY significant difference will show that some patients developed problems causally due to statins. Use of a 1cm threshold, after enrolling a sample of uncertain causality, appears to reflect a determination not to show an effect. And the sample size should be inflated to consider your perspective that not all prior reports need have been causal.

“Using simulation, we estimated a sample size...” – with what inputs? On what basis? Be sure to analyze dropouts as a critical outcome. Participants who drop in the first period – or in the second period prior to VAS – are not considered. Please include simple analysis to show initial effects:

- Change in mean score from no treatment onto statin, and from no treatment onto placebo
- Change in mean score from first treatment phase to second: placebo to statin; and from statin to placebo
- Then mean of change in mean, for each statin-to-placebo, and each placebo-to-statin
- Dropout in first phase if placed on statin; dropout in first phase if placed on placebo
- Dropout in first statin phase, whichever time; compared to dropout in first placebo phase
- Inquire about cause of dropout and analyze

Analyze the percent who develop symptoms overall in addition to the percent that attribute to statins on statin vs placebo. These can and should be a simple analysis. Separately analyze the first and second phases as longer use may lead to longer recovery time and more difficulty distinguishing statin from placebo. Please clarify what this means. “We will investigate whether the excess muscle symptoms, if any, experienced during treatment periods with the statin appears to be concentrated in multiple sites.” “concentrated” and “multiple” are a confusing pair of words.

Discussion

Limitations: Self-selection was partially addressed. However, potential for extreme self-selection should be more fully acknowledged. Patients who have strong cause to believe their problem was due to a statin will be justifiably disinclined to rechallenge. Those whose problems, legitimately due to statins, were persistent may be justifiably disinclined to participate. Participants may (indeed should) primarily be those whose problem was equivocally present, and/or who aren't persuaded the problem was necessarily due to the statin. Thus it will assess the subgroup in whom a causal relationship is least likely, and for whom reexposure to statins may least likely produce problems. Generalizations will be made from these findings to patients whose problems are more likely related to statins, and more serious – in whom the peril of rechallenge will be potentially far greater. This limitation should be fully acknowledged; it should be made clear that findings for these patients cannot be thought to generalize to other patients who reported muscle problems on statin, and did not elect to participate. Citations: The citations are weighted strongly toward articles that disclaim most statin adverse effects – and the present article appears to accept such characterizations.

These include citation of the 2016 Lancet “review” by Collins et al⁴⁰, on which the corresponding author for this paper was a coauthor; and the article by Finegold⁴¹ et al on which another of the investigators participated (none of the considerations exposed in a rebuttal to that article are reflected¹⁵). A further citation, with first author Armitage⁴² (an investigator on this study), states that even serious muscle adverse effects/ rhabdomyolysis usually reverse fully. Based on what? This does not comport with our experience. As Dr. Phillips noted, persistence of weakness was more common in rhabdomyolysis than in non-CK elevating statin muscle problems; but even in the latter, in our published data on non-rhabdomyolysis statin muscle problems, a minority reported that recovery was complete². As Dr. Phillips, with his statin myopathy clinic, noted, the fact that a patient stopped complaining (or showing up) does not imply their recovery was complete³⁰. And Dr. Vladutiu noted, in her study examining genetic predisposition to statin muscle problems, “Variable persistent symptoms occurred in 68% of patients despite cessation of therapy”⁸. Other citations such as those of Barsky and Zhang fit the slant of the citations. There is currently no meaningful reflection (or citation) of relevant materials providing contrary data and inference, such as those documenting mechanism (and concurrent findings like fatty acid oxidation and respiratory exchange ratio changes), disability, and dose-response – or symptom persistence^{1, 6-9, 25, 30, 43}; supporting the presence of adverse effects (including by documenting risk factors for them), or challenging the perspectives of the cited documents^{9, 14, 15}. To allow an appearance of impartiality on the issue, or to allow the perception that this is an honest effort to examine the purported topic of this study, the authors need to make an aggressive effort to portray both knowledge of evidence on the other side, and an actual openness to the possibility that statins may causally produce problems they purport to be studying. Ideally they should incorporate individuals in their investigator group who are experts on these statin muscle problems (such as Dr. Paul Phillips, Dr. Georgirene Vladutiu, Dr. Helmut Sinzinger, and/or Dr. P. Thompson), if any are ethically comfortable with participating. This would assist in development of a design more advantageous to detecting problems if they occur; and to negate the appearance that the study’s purpose, based on past articles by the authors, and the selection of citations, may be to not find problems.

Statements needing revision

“Clinicians faced with patients presenting with muscle symptoms during statin use are encouraged to measure the blood creatine kinase, which is substantially elevated in the rare cases of statin associated myopathy, but in the vast majority of patients the CK level will be normal, ruling out myopathy.” {emphasis mine} - This statement suggests ignorance of the evidence (or distortion). Data by Phillips^{1, 30} (and others) clearly show that non-CK elevating muscle symptoms on statin can absolutely represent myopathy.

The reconstruing of myopathy, in some publications, to require high CK elevation (with an arbitrary cutoff) to report myopathy on statins, seems clearly designed to reduce acknowledgment of problems.

“Strengths

“This trial will determine whether statins cause muscle symptoms during statin use in these participants, allowing clinician and participant to make an informed decision about continued use.”

- It will only determine whether atorvastatin at 20mg given for 2 months (with 2 months allowance for recovery), does so, in patients who may only have had problems on higher potency agents.

“This trial addresses some of the limitations of previous statin side effects literature.”

- Toward what purpose? Which limitations? There is already a blinded placebo-controlled dechallenge-rechallenge study, that goes farther than this by adding biopsy assessment; this and other data eliminate doubt about the existence of statin muscle adverse effects. This study has different significant limitations, which should be better acknowledged. “In these cases there is currently no other diagnostic tool allowing clinicians to empirically evaluate whether symptoms reported by an individual statin-user are caused by the statin itself or by the ‘nocebo’ effect.”

- Since the nocebo effect has less evidence than causal statin muscle effects (reviewed below), this statement needs to be revised. The necessity and burden of a placebo dechallenge is likely to be seldom necessary, particularly given the evidence of quite modest expected benefits of statins³². The first approach is to reevaluate how compelling is the need for the drug. Then, if necessary, a trial of dechallenge-rechallenge could be attempted.

“Randomised controlled blinded trials (RCTs) have found no evidence of an effect on these symptoms.^{12}”

- As repeatedly reported, because of effect modification, average effects from a randomized trial that fail to show significant harm to an outcome in the treatment group, have little bearing on whether the drug can cause that harm in a vulnerable subgroup. This has been shown across many treatments – bisphosphonates can cause fracture; anxiolytics can cause anxiety; antidepressants can cause suicide; antihypertensives can “paradoxically” raise blood pressure, as can salt restriction. Antioxidants can almost always have prooxidant effects, and this likely underlies statins’ bidirectional effects on many outcomes⁹.

- Statins may reduce risk of muscle pain in some, by improving energy and oxidative stress, just as it contributes to them, impairing energy and worsening oxidative stress in others⁹. That some may benefit on the drug, does not reduce the impact or drug contribution for those who are harmed. Analogously, statins reduced diabetes in the WOSCOPS trial⁴⁴ but increase it in studies with high potency or participants with risk factors for statin problems – e.g older age or metabolic syndrome factors (which are also risk factors for diabetes).

Moreover, in some studies (like JUPITER), neutral average effects on glucose accompany increases in incident diabetes, comporting with bidirectional effects. The only means by which statin muscle problems can be challenged, is via methodological failure – failure to recognize and understand the issue of “effect modification,” which causes bidirectional effects with many drugs and exposures (and particularly ones that have antioxidant effects, which typically provides prospects for prooxidant effects)⁴⁵.

- Additionally, that statement is untrue. E.g.:

- o Although this has been published only as an abstract, it was shown clearly that by coupling known risk factors to identify a high risk group within a randomized trial (because, again, of recognition of the existence of effect modification), such as female sex and older age, muscle weakness was significantly greater on statin than on placebo. This was separately significant for simvastatin vs placebo and for pravastatin vs placebo, so is not plausibly attributable to chance⁴⁶.
- o A significant increase in fatigue was also found in the full sample for that RCT⁴⁷. The article reports on a composite measure (fatigue and exertional fatigue) because this was demanded by reviewers; but the original submission showed significance separately for the two fatigue-related measures, so that concerns based on composite measures do not apply.
- o And one of the participants on this WISE effort was a coauthor on the "Search" trial. What of the evidence procured on what was termed "incipient myopathy"? (really, two-organ, combined muscle and liver adverse effects – but with muscle problems short of rhabdomyolysis).
- o There is also the Silver 2006 meta-analysis of RCTs. "Statin therapy increased the risk of any AE by 39% (OR = 1.4; 95% CI, 1.09-1.80; P = 0.008.... Nonurgent AEs such as myalgia and liver function elevations were responsible for approximately two thirds of AEs reported in trials."

- Rhabdomyolysis has also generally not shown up as a statin AE in RCTs and meta-analyses of them:
 Why are other forms of evidence accepted for rhabdomyolysis, but not muscle problems short of this?
 "A recent review of the evidence from randomised trials and observational studies suggested that symptomatic adverse events may be misattributed to statins.^{9}"

- The authors elected to ignore the issue of effect modification, though the importance of this methodological consideration in relation to statin AEs has been raised repeatedly^{9, 14, 15}, including in a published response to a similar claim¹⁵. "Uncertainty about the association between muscle symptoms and statins persists due to limitations of observational studies and trials."
- Any uncertainty necessitates either inadequate knowledge of the literature, or inadequate methodological aptitude.

"If the CK and ALT are within the normal range and the participant remains eligible they can be offered the following options by the GP:"
 What exactly is the plan if they are abnormal?
 Is there no concern that these may be very high before the participant cites symptoms, in someone selected for potentially being at risk?
 "A key secondary outcome is pain for which the participant answers "yes" or "don't know" to the question "Do you think these muscle symptoms are related to your study medication?" asked at the end of each treatment period. People answering "No" to this question would have their VAS scores set to zero for that treatment period for this specific secondary outcome."

- Please be more clear about whether this "zero" just refers to "no" (don't think it is related), or refers to the outcome of pain?

What is the rationale for looking at genetic predictors of statin effects if you are not persuaded these exist?

“A major limitation of observational studies is a lack of blinding; patients taking a medication expect to experience adverse effects^{13} and therefore reporting of symptoms in statin users may be higher than in a comparable population not on statins. This phenomenon, the ‘nocebo’ effect, can lead to bias in unblinded studies.”

- This needs to be revised based on the actual evidence regarding placebo-nocebo effects. Unlike statin adverse effects (wherein effect modification renders average effects of RCTs irrelevant to subsets), the whole premise of placebo effects is the idea that on a placebo, randomized against an active drug, expectation produces average differences in reporting relative to if there had been no treatment. This is empirically testable, and the determination should rely on controlled trials. Controlled studies, reviewed by the Nordic Cochrane Center, in fact show little evidence for any placebo effects (Nordic Cochrane Center data)^{23, 48}: In trials that include a randomized “no treatment” arm together with a randomized placebo arm, evidence fails to validate net placebo effects except for analgesic effects (which are shown in a separate literature to be mediated by release of endogenous opiates, and blocked by naloxone – i.e. their mechanism that does not generalize to other treatments and outcomes). What people have attributed to placebo/nocebo effects derive from other considerations (including, in some cases, active effects of the placebo⁴⁹⁻⁵¹.)

To summarize, this has potential to be a useful study, and there exists the prospect of an acceptable manuscript in relation to this. For this to be the case, the manuscript must better reflect and acknowledge existing evidence that documents existence of statin muscle adverse effects, including effects short of rhabdomyolysis. It must characterize its purpose differently; it is not ethical to expose patients to a drug, who may have already responded to that drug adversely, for the purpose of showing something that is already known. It must revise the design, capitalizing on existing information to better guide dose and duration, in order to have ability to assess reproducibility of adverse muscle effects arising on statins; it must better acknowledge the uncertainties in risks and benefits that may redound to the patient, both in the article and to prospective participants themselves. It must more openly characterize its limitations.

References

1. Phillips PS, Haas RH, Bannykh S, et al. Statin-associated myopathy with normal creatine kinase levels. *Ann Intern Med* 2002;137:581-5.
2. Cham S, Evans MA, Denenberg JO, Golomb BA. Statin-associated muscle-related adverse effects: a case series of 354 patients. *Pharmacotherapy* 2010;30:541-53.
3. Hoffman KB, Kraus C, Dimbil M, Golomb BA. A Survey of the FDA's AERS Database Regarding Muscle and Tendon Adverse Events Linked to the Statin Drug Class. *PLoS One* 2012;7:e42866.
4. Dale KM, Coleman CI, Henyan NN, Kluger J, White CM. Does More Aggressive Statin Therapy Increase Muscle and Liver Risk? In: 55th Annual Scientific Sessions of the American College of Cardiology; 2006 Mar 11-14; Atlanta, GA; 2006.
5. Link E, Parish S, Armitage J, et al. SLCO1B1 variants and statin-induced myopathy--a genomewide study.

- N Engl J Med 2008;359:789-99.
6. Sinzinger H, Lupattelli G, Chehne F, Oguogho A, Furberg CD. Isoprostane 8-epi-PGF₂α is frequently increased in patients with muscle pain and/or CK-elevation after HMG-Co-enzyme-A-reductase inhibitor therapy. *J Clin Pharm Ther* 2001;26:303-10.
 7. Phillips PS, Ciaraldi TP, Kim DL, Verity MA, Wolfson T, Henry RR. Myotoxic reactions to lipid-lowering therapy are associated with altered oxidation of fatty acids. *Endocrine* 2009;35:38-46.
 8. Vladutiu GD, Simmons Z, Isackson PJ, et al. Genetic risk factors associated with lipid-lowering drug-induced myopathies. *Muscle Nerve* 2006;34:153-62.
 9. Golomb BA, Evans MA. Statin adverse effects : a review of the literature and evidence for a mitochondrial mechanism. *Am J Cardiovasc Drugs* 2008;8:373-418.
 10. Mammen AL, Chung T, Christopher-Stine L, et al. Autoantibodies against 3-hydroxy-3-methylglutarylcoenzyme A reductase in patients with statin-associated autoimmune myopathy. *Arthritis Rheum* 2011;63:713-21.
 11. Grable-Esposito P, Katzberg HD, Greenberg SA, Srinivasan J, Katz J, Amato AA. Immune-mediated necrotizing myopathy associated with statins. *Muscle Nerve* 2010;41:185-90.
 12. Liang C, Needham M. Necrotizing autoimmune myopathy. *Curr Opin Rheumatol* 2011;23:612-9.
 13. Barbiroli B, Frassinetti C, Martinelli P, et al. Coenzyme Q10 improves mitochondrial respiration in patients with mitochondrial cytopathies. An in vivo study on brain and skeletal muscle by phosphorous magnetic resonance spectroscopy. *Cell Mol Biol (Noisy-le-grand)* 1997;43:741-9.
 14. Golomb BA. The importance of monitoring adverse effects in statin, and other, clinical trials. *Clin Invest* 2013;3:913-6.
 15. Golomb BA. Misinterpretation of trial evidence on statin adverse effects may harm patients. *Eur J Prev Cardiol* 2015;22:492-3.
 16. Turner EH, Knoepflmacher D, Shapley L. Publication bias in antipsychotic trials: an analysis of efficacy comparing the published literature to the US Food and Drug Administration database. *PLoS Med* 2012;9:e1001189.
 17. Stelfox HT, Chua G, O'Rourke K, Detsky AS. Conflict of interest in the debate over calcium-channel antagonists. *N Engl J Med* 1998;338:101-6.
 18. Bero L, Oostvogel F, Bacchetti P, Lee K. Factors Associated with Findings of Published Trials of Drug-Drug Comparisons: Why Some Statins Appear More Efficacious than Others. *PLoS Med* 2007;4:e184.
 19. Barnes DE, Bero LA. Why review articles on the health effects of passive smoking reach different conclusions. *Jama* 1998;279:1566-70.
 20. Harris G. Caustic Government Report Deals Blow to Diabetes Drug. *New York Times* 2010;July 9, 2010.
 21. Did GSK trial data mask Paxil suicide risk? *The New Scientist* 2008; 8 February 2008 12.
 22. Smith R. Conflicts of interest: how money clouds objectivity. *J R Soc Med* 2006;99:292-7.
 23. Hrobjartsson A, Gotzsche PC. Placebo treatment versus no treatment. *Cochrane Database Syst Rev* 2003:CD003974.

24. Gaist D, Jeppesen M, Andersen LA, Garcia Rodriguez J, Hallas J, Sindrup SH. Statins and risk of polyneuropathy: A case-control study. *Neurology* 2002;58:1333-7.
25. Phillips PS, Phillips CT, Sullivan MJ, Naviaux RK, Haas RH. Statin myotoxicity is associated with changes in the cardiopulmonary function. *Atherosclerosis* 2004;177:183-8.
26. Mohaupt MG, Karas RH, Babiychuk EB, et al. Association between statin-associated myopathy and skeletal muscle damage. *CMAJ* 2009;181:E11-8.
27. Phillips CT, Gray NL, Puhek LM, McDonald FG, Sullivan MJ, Phillips PS. Cardiopulmonary Function Is Abnormal After Statin-Induced Rhabdomyolysis and Myositis [abstract]. *J Am Coll Cardiol* 2004;43 (suppl A):16a.
28. Phillips CT, Gray NL, Puhek LM, McDonald FG, Sullivan MJ, Phillips PS. Basal Respiratory Exchange Ratio Is Altered With Statin Use in Normals [abstract]. *J Am Coll Cardiol* 2004;43 (suppl A):233a.
29. Phan T, McLeod JG, Pollard JD, Peiris O, Rohan A, Halpern JP. Peripheral neuropathy associated with simvastatin. *Journal of Neurology, Neurosurgery and Psychiatry* 1995;58:625-8.
30. Phillips PS, Haas RH. Observations from a statin myopathy clinic. *Arch Intern Med* 2006;166:1232-3.
31. Koslik HJ, Meskimen AH, Golomb BA. Physicians' Experiences as Patients with Statin Side Effects: A Case Series. *Drug Safety - Case Reports* 2017;4:3.
32. Kristensen M, Christensen P, Hallas J. The effect of statins on average survival in randomised trials, an analysis of endpoint postponement. *BMJ Open* 2015;5:e007118.
33. Graham DJ, Staffa JA, Shatin D, et al. Incidence of hospitalized rhabdomyolysis in patients treated with lipid-lowering drugs. *Jama* 2004;292:2585-90.
34. Camargos EF, Oliveira Lde F, Boaventura Tde D. My legs are getting old: simvastatin-induced polyneuropathy. *BMJ Case Rep* 2011;2011.
35. Jeppesen U, Gaist D, Smith T, Sindrup SH. Statins and peripheral neuropathy. *Eur J Clin Pharmacol* 1999;54:835-8.
36. de Langen JJ, van Puijenbroek EP. HMG-CoA-reductase inhibitors and neuropathy: reports to the Netherlands Pharmacovigilance Centre. *Neth J Med* 2006;64:334-8.
37. Backes JM, Howard PA. Association of HMG-CoA reductase inhibitors with neuropathy. *Ann Pharmacother* 2003;37:274-8.
38. Adverse Drug Reactions Advisory Committee (ADRAC). Statins and peripheral neuropathy. *Australian Adverse Drug Reactions Bulletin* 2005;24:6.
39. Otruba P, Kanovsky P, Hlustik P. Treatment with statins and involvement of the peripheral nervous system: results of a prospective clinical and neurophysiological follow-up. *Biomed Pap Med Fac Univ Palacky Olomouc Czech Repub* 2007;151:307-10.
40. Collins R, Emberson J, Armitage J, et al. Interpretation of the evidence for the efficacy and safety of statin therapy. *TheLancetcom* 2016;Published Online September 8, 2016 [http://dx.doi.org/10.1016/S0140-6736\(16\)31357-5](http://dx.doi.org/10.1016/S0140-6736(16)31357-5).
41. Finegold JA, Manisty CH, Goldacre B, Barron AJ, Francis DP. What proportion of symptomatic side effects in patients taking statins are genuinely caused by the drug? Systematic review of randomized placebo-controlled trials to aid individual patient choice.

	Eur J Prev Cardiol 2014;21:464-74. 42. Armitage J. The safety of statins in clinical practice. Lancet 2007. 43. Vladutiu GD. Genetic predisposition to statin myopathy. Curr Opin Rheumatol 2008;20:648-55. 44. Freeman DJ, Norrie J, Sattar N, et al. Pravastatin and the development of diabetes mellitus: evidence for a protective treatment effect in the West of Scotland Coronary Prevention Study. Circulation 2001;103:357-62. 45. Golomb B. Effect modification. In: Edge.org, ed. What Scientific Term or Concept Ought to Be More Widely Known; 2017. 46. Golomb BA, Koperski S. Who becomes weak on statins? Effect modification exposed in a RCT by risk factor compounding. Circulation 2013;127:AP072. 47. Golomb BA, Evans MA, Dimsdale JE, White HL. Effects of statins on energy and fatigue with exertion: results from a randomized controlled trial. Arch Intern Med 2012;172:1180-2. 48. Hrobjartsson A, Gotzsche PC. [What is the effect of placebo interventions? A systematic review of randomized clinical trials with placebo treated and untreated patients]. Ugeskr Laeger 2002;164:329-33. 49. Golomb B. When are medication side effects due to the nocebo phenomenon? Jama 2002;287:2502-3; discussion 3-4. 50. Golomb BA. Paradox of placebo effect. Nature 1995;375:530. 51. Golomb BA. Are placebos inert or powerful? Vice versa. Clinical Investigation 2011;1:1471-3.
--	---

VERSION 1 – AUTHOR RESPONSE

Reviewer: 1

Reviewer Name: Gabriel Chodick

Institution and Country: Maccabi Institute for Research & Innovation, Tel-Aviv University, Israel

Competing Interests: None

This is a well-written study proposal of N-of-1 trial aimed to assess the relationship between muscle symptoms and statins as well as patients' perceptions regarding this relationship. N-of-1 trial is a most effective way to address this issue with minimal biases and therefore, this publication might be of interest for readers. The planned study seems to be well powered for VAS difference of at least 1cm, both by number of patients and also by the number of treatment periods.

Comment 1. The study design is heavily dependent on the assumption that statins-related muscle symptoms are relived shortly after statins are discontinued and re-experienced if therapy is reinitiated. This might be the case; but more text is required on this strong assumption.

Response: We thank the reviewer for highlighting this important issue. Our design was supported by previous work in this area, which we have now cited in the manuscript (page 10)

Comment 2. Adherence with therapy is a major challenge in this population; did the authors consider measuring LDL levels at different periods to validate drug consumption? To keep blinding, patients should be allowed to get their LDL levels while under therapy.

Response: This is a nice idea to get an idea of adherence to the study drugs. However, our aim was to minimise the burden on participants and GPs, and to keep the trial as pragmatic as possible. We request that participants report their adherence to the drug (daily yes/no in the last seven days of each treatment period, and also in the questionnaire at the end of each treatment period we ask for a qualitative overview of use over the previous two months), and also return the medication blister packs to the trial team using prepaid envelopes, allowing us to assess how many tablets have been used in each period. Therefore we will ascertain a measure of adherence.

Comment 3. It is unclear whether a washout period is planned at study initiation phase.

Response: We thank the reviewer for highlighting the lack of clarity here. There is no washout period at study initiation and we have now clarified this in the manuscript (page 9). However, in each treatment period on either statin or placebo, participants will have been on the relevant treatment (and hence off the prior treatment) for seven weeks before any symptom scores are collected.

Comment 4. Patient safety: pregnancy issue should be explicitly mentioned Pregnant women are excluded from the trial on the basis that atorvastatin is contraindicated in this group.

Response: We have now explained that pregnant women are excluded on the basis that statins are contraindicated in this group (page 8).

Comment 5. Feasibility: N-of-1 trials are complex and may burden on the caregivers and research team. More details are required on how this should be effectively managed.

Response: This trial has been designed with the patient and caregiver in mind, being as simple as possible. The burden on GPs has been minimised through the use of research nurses for the bulk of the work. The burden on patients has been minimised through postage of study drugs direct to patients homes, and the choice of data collection methods. We have also conducted patient and public involvement to help with the trial design and data collection methods. We have added some brief text on page 7 to fully explain this aspect.

Reviewer: 2

Reviewer Name: Johann Auer

Institution and Country: Department of Cardiology and Intensive Care, St. Josef Hospital Braunau, Austria

Competing Interests: none

This study will investigate 200 patients in a randomised, double blinded N-of-1 design in primary care and will determine (i) the effect of statins on all muscle symptoms, and (ii) the effect of statins on muscle pain that is perceived to be statin related. Patients who are considering discontinuing statin use due to muscle symptoms, and those who have discontinued in the last three years due to such symptoms, will be recruited. Participants will be randomised to a sequence of six two-month treatment periods during which they will receive atorvastatin 20mg daily or matched placebo. On each of the last seven days of each treatment period, participants will rate their muscle symptoms on a Visual Analogue Scale (VAS).

This issue is timely and the scientific question is worth to be investigated. The authors use a novel web-based setting. Participants will choose the method of data collection most suitable for them.

In general, the design is appropriate and seems to be scientifically accurate. I have some minor points that should be addressed:

Comment: The authors state that this is a „....series of randomised controlled ... trials...“. This may suggest that this is not one but more trials. However, this is one trial that included 200 patients in a N-of-1 design. This should be clarified to avoid confusion.

Response: Each person in our study can be seen as having their own RCT, and therefore the StatinWISE study is a series of 200 N-of-1 RCTs, equivalent to a single multiple-crossover study of 200 patients. We have clarified this point in the manuscript on page 7.

Comment: The authors focus on muscle symptoms. However, statins are associated with other skeletal-related symptoms beyond muscles. Assessment of such symptoms would give further insights into this issue.

Response: StatinWISE has been designed to minimise burden on participants and GPs. One of the ways we've done this is to look at a focused question, rather than trying to elucidate information about all possible side effects.

Comment: Blinding may be compromised by (unscheduled) LDL-C. measurements (driven by the patient) during the study. How do the authors handle this issue?

Response: While unscheduled LDL-C measurements are possible, they are unusual in UK primary care. To compromise blinding, the patient and/or clinician would need to deliberately relate the result to pre-trial measurements and to treatment status at the time of the previous measurements. While this is not impossible, we doubt it will have any major impact on blinding in this study.

Reviewer: 3

Reviewer Name: Beatrice A. Golomb, MD, PhD

Institution and Country: University of California, San Diego, United States

Competing Interests: None declared.

We would like to thank the reviewer for her considerable effort in reviewing our manuscript. Below, we have addressed some of the key themes raised by the reviewer. Then, in the pages that follow, we have separately addressed the specific comments.

Many of the comments appear to stem from the referee's individual opinions about statins, their benefits, and their adverse effects. While we fully respect her right to hold such opinions, where these are based on personal preferences and judgements rather than on the scientific evidence, there is limited scope for a constructive response from us other than pointing out we disagree.

The empirical scientific evidence to date shows that we do not know how often statins cause muscle symptoms: hence the need for this study.

Comment: Characterization of evidence regarding muscle adverse effects

Response: The key issue raised by the reviewer was the lack of cited evidence on adverse effects and their potential mechanisms.

Comment: We previously summarised high-quality randomised evidence regarding this relationship in our Introduction, and there is little such evidence that supports the existence of an association, leading to the paucity of cited evidence in that direction in our original manuscript.

We have now added some of the studies highlighted by the reviewer, in order to demonstrate to readers that there are a range of opinions on the issue. However, we have omitted suggested articles that we feel have a higher chance of their results being due to chance or bias.

Responses: As stated above, the empirical scientific evidence to date shows that we do not know how often statins cause muscle symptoms: hence the need for this study.

Design

The reviewer has made a number of suggested alterations to the trial protocol. We welcome suggestions for points that we might raise in the discussion about aspects of the design and possible limitations of the study. However, the StatinWISE trial was scientifically peer reviewed prior to being funded by the UK NIHR and has been reviewed and approved by both the UK-MHRA and Ethics Committee. Our aim in publishing the protocol is to make the study as accessible and transparent as possible and to increase the visibility and level of detail available for people interested. Therefore, in submitting the protocol for publication, we are not able to address issues relating to the design of the study and cannot make changes to this.

Importantly, the main aim of StatinWISE is to determine whether muscle symptoms during statin use are caused by statins in particular individuals. The stated limitations of our design (page 4 of our protocol) and those raised by the reviewer reflect the trade-off between feasibility and answering a useful and important study question. Our trial is designed to address one specific question rather than addressing a large number of questions simultaneously. We have now clarified our aim on page 6 and stated on page 4 the limitation of the study assessing a single statin at a single dose (which it is worth noting is the most commonly used statin at the most commonly used dose).

Analyses

Response: We thank the reviewer for the suggestions regarding our analysis. We will report our analyses according to the CONSORT extension statement for N-of-1 trials.

Comment: The authors cite desirability of N-of-1 trials of statins in those with perceived statin symptoms for the purpose of determining whether muscle symptoms short of rhabdomyolysis can be caused by statins. Strengths of the study include use of N-of-one randomized double blind design. Another strength is the inclusion of a question whether the patient believes the symptom to be statin-related. Based on data in an RCT we conducted, ~40% on placebo thought they were on statin, ~60% on statin did so – if the difference is similar, a larger sample may be needed. In principle, a properly conducted study of this kind could yield important information. And a write-up of such a study could be rendered acceptable. However, in its current form the manuscript (and accompanying study design) have significant limitations.

Attention to these is necessary for the paper to be made suitable for publication – and for the underlying study to be scientifically and ethically viable. (Members of the ethics review board were presumably not experts on this issue, and were guided by the presentation of evidence – including omissions therein.)

Comment 1. Characterization of evidence regarding muscle adverse effects: The presence of such multiple lines of evidence affirming the existence of statin muscle problems (short of rhabdomyolysis) should be fully acknowledged.

Evidence in support of this has already included (among other findings): double-blind N-of-one dechallenge-rechallenge biopsy evidence¹; unblinded dechallenge-rechallenge from which potency/dose-response evidence naturally emerged – in patients highly unlikely to be aware of potency relationships²; emergent potency evidence in FDA AERS data that matches the findings from the aforementioned dechallenge rechallenge data, and documents elevated relative odds ratios of reporting for muscle problems relative to other drugs³. RCT comparisons of higher-vs-lower statin potency further support dose dependence in CK elevations⁴. RCT evidence in which a gene variant, particularly coupled with higher potency statins, was linked not only to “myopathy” (rhabdomyolysis) but to what was termed “incipient myopathy” which was really two-organ problems encompassing muscle and liver dysfunction⁵ – but, encompassing muscle. One of the participants on this was an author on that study, which should not have been omitted.

Response: We thank the reviewer for highlighting these papers and we have the following response:

- REF 1: This is a study describing four case reports. The RCT from which they came is not cited in the article. We are unable to find any subsequent published RCT, so it is unclear what the umbrella RCT design was, or what its results were.
- REF 2: We have now cited this work in justifying our two month time periods. However, it was a case series of 354 patients, all exposed to statins and self-selected patients based on perceived statin associated adverse effects. The lack of blinding and comparison group in this study limits its findings.
- REF 3: This is based on data from the FDA’s Adverse Event Reporting System, so is again non-randomised and unblinded data. Findings may possibly have been influenced by publicity of muscle issues, since patients and practitioners were unblinded.
- REF 4: This is a conference abstract. There is no evidence of the RCT being later published and we are unable to cite this.
- REF 5: This is a genome wide association study looking at people with myopathy (either CK >10 times upper limit of normal, or CK > 3 times upper limit of normal AND >5 times baseline value. We are aware that there are genetic variants associated with myopathy, but this is a more severe symptom than the ones we are investigating in StatinWISE) and therefore have not cited this paper. We have clearly stated that statins are associated with myopathy on page 5 (Introduction). Of these citations, none provides randomised evidence of the statin-muscle-symptom (except myopathy) association. However, we have now cited References 2 and 3 in the manuscript to broaden the opinion base we report.

Comment 2. Evidence has already documented mechanisms associated with (non-rhabdomyolysis) muscle problems due to statins, including oxidative stress (via statins’ prooxidant antioxidant duality⁶), mitochondrial/ metabolic mechanisms/ fatty acid oxidation abnormalities^{1, 7-9}, and autoimmune necrotizing myopathy¹⁰⁻¹². Genetic predispositions of a character that supports statin induction of metabolic myopathies has also been reported⁸. Additionally, dose-dependent withdrawal of coenzyme Q10 with statins (reducing coQ10 in parallel with the cholesterol on which it is transported, since both are products of the mevalonate pathway) is expected to unmask mitochondrial pathology⁹, and thus, in a subset, muscle symptoms, just as supplementation with coenzyme Q10 can relatively bypass any of a range of mitochondrial abnormalities¹³. The existence of such evidence should be fully acknowledged.

The relevance of the suggested strands of evidence to muscle symptoms (as opposed to serious myopathy) is tenuous and not of direct relevance to our trial.

As stated above, the empirical scientific evidence to date shows that we do not know how often statins cause muscle symptoms: hence the need for this study.

We suggest that including speculation on possible mechanisms would not be helpful in a trial protocol.

Comment 3. Based on the above, clearly, muscle symptoms other than rhabdomyolysis do occur, due to statins, and in any given patient, muscle symptoms arising on statins, and perhaps particularly if attributed by the patient to statins, may be due to the statin. If this study shows low rates of recurrence (even if the dose and duration issues are fixed), it will not inform us about those patients, because we cannot know how the character of patients who participate relates to the character of patients who do not.

Response: We disagree with assertions in points 3 and 4. Given the evidence from large systematic reviews and meta-analyses, we believe that the existence of statin muscle-related side effects, other than rhabdomyolysis and myopathy, is still unclear, hence our justification for the trial. We agree that this trial will not tell us anything about the people who do not participate. The reviewer suggests mechanistic pathways which may explain non rhabdomyolysis pain. However, this is not supported by evidence from large systematic reviews and meta-analyses.

Comment 4. Analogous to the case of glucose, muscle effects of statins are linked to oxidative stress and thus bidirectional changes may produce neutral net effects on statin relative to placebo – even when high dose or multiple risk factors increase risk. This point has been made elsewhere^{14, 15}. Whether statin problems are or are not due to the statin in other patients, does not obviate the possibility that statins causally produced the problem in this patient. This should be acknowledged.

Response: Our intention in this trial is to ascertain whether any individual participant has muscle symptoms that are caused by statins. It is for this reason that the N-of-1 design is appropriate for this question – it allows us to ascertain individualised results. 200 of these individual results will allow us to infer beyond the individuals: such more generalised inference will be clear and explicit.

Comment 5. Copious evidence shows a powerful relationship of conflict of interest to study outcome and interpretation¹⁶⁻²¹. As former BMJ editor-in-chief Richard Smith noted, in relation to findings of a study, “It suggests that, far from conflict of interest being unimportant in the objective and pure world of science where method and the quality of data is everything, it is the main factor determining the result of studies”²².

In this vein, it is of concern that articles are primarily cited that, unless they address other points, seek to repudiate a statin relationship to muscle symptoms. This concern is magnified by the prior involvement of several of the authors/ investigators here, in articles that dismiss the existence of causal statin effects on muscle. In this context it is imperative that the authors also demonstrate that they are familiar with, and can represent, the best case for the contrary view. Otherwise there is the appearance that the study may be designed and implemented by parties – who have previously sought to discredit statin muscle adverse effects – potentially for the purpose of further discrediting their existence, if such effects do occur. And with any honest reading of the literature, coupled with basic methodological acumen, they do occur. Therefore I suggest the purpose be recast as assessing how frequently instances can be validated, among persons amenable to rechallenging, with the present design. (Considerations below of dose and duration still require attention.)

As stated above, the empirical scientific evidence to date shows that we do not know how often statins cause muscle symptoms: hence the need for this study.

Response: It was not our intention to repudiate a statin relationship to muscle symptoms. Instead, we aimed to represent the totality of the evidence by citing major systematic reviews and meta-analyses. We are aware that this evidence suggests that there is no effect of statins on muscle symptoms. However, our Introduction states that patients continue to believe that their symptoms are statin-related. We have now added citations from some of these smaller and largely unblinded studies suggested by the reviewer to demonstrate the opinions that some people hold on the issue.

However, the justification for our trial remains the same: there is scientific uncertainty regarding whether statins cause muscle symptoms.

Comment 6. It is also vital that the design of the study is up to snuff, that it have the ability to detect muscle problems if they arise. If not, such a study may be frankly damaging, by fostering a false narrative and exposing additional patients to risk and morbidity. Several modifications to the design are in order.

Response: As stated above, we welcome suggestions of possible limitations for discussion, but we are not in a position to make changes to the study design.

Design

a. Dose: The authors must acknowledge that dose-potency is an issue. And they must adjust the design accordingly. Evidence for dose/ potency dependence of statin adverse effects (previously reviewed⁹), renders the present design problematic. Persons with muscle symptoms on statins who had discontinued the statin, then were rechallenged by their physician (unblinded) with a statin for at least two months, reliably reexperienced the problem – if the rechallenge statin was of higher expected potency than the original statin². The sample was self-selected. It is possible that those who did not reexperience the problem may have been less likely to respond. However, that would not account for the dose relationship found. This is unlikely to reflect expectation/ nocebo effects, not only because according to the Nordic Cochrane center such effects – in a controlled setting, largely do not exist (placebo is generally not different from no treatment, when there is randomization to both²³); but because even if they did, most physicians are/ were not closely familiar with potency equivalencies, much less were patients. The recurrence rate was still substantial with lower statin potency (over 70%), at least among those who submitted adverse event reports to the patient-targeted database.

However in the modern era there is greater use of high potency statins – and it cannot be presumed that persons will experience statin problems on atorvastatin 20mg, who previously experienced problems on 80 or 40mg, or on rosuvastatin 40mg (or 20mg). So for the stated purpose, to examine whether statin problems are real, rather than whether there is a statin dose which patients can tolerate, it is important that an adequate potency statin be employed. (Though as above, the purpose needs to be revised.)

{Of note, “equivalent potency” but different statin agents may be problematic. First, potency equivalencies are inexact. Second, persons may have genetic or other factors (renal, hepatic, grapefruit juice, concurrent interacting medications) that may fluctuate over time, or that may affect clearance of one but not another statin, producing a higher (or lower) effective potency of the index statin than would be reflected in the subsequent “equivalent” potency statin. Changes in nutritional status, organ function, or concurrent oxidant exposures could influence experience of the problem differentially with reexposure (less, or more), producing different results. Such factors are less likely to be decisive when a higher potency agent is used for rechallenge. (Higher potencies may pose risk of greater problems – of course, that is for those of us who acknowledge the existence of such problems.) The design must adequately consider this. It would clearly be unethical to conduct a study of this kind in a fashion that might disrespect patients’ contribution by subjecting them to risk, while missing a real problem. Therefore, the study design must be revisited to better accommodate dose and duration considerations.

With respect to the comments about the nocebo effect;

We cannot dismiss the nocebo effect. The reviewer’s citation about the non-existence of the placebo effect misrepresents the conclusions of the cited Cochrane study.

The main results of the study were: “There was no statistically significant effect of placebo interventions in eight out of nine clinical conditions investigated...”

There was a modest apparent analgesic effect of placebo interventions, standardised mean difference -0.27 (-0.40 to -0.15), but also a substantial risk of bias.”

Their overall conclusion was “There was no evidence that placebo interventions in general have clinically important effects. A possible moderate effect on subjective continuous outcomes, especially pain, could not be clearly distinguished from bias.” Claiming that this review shows the placebo effect does not occur for pain outcomes misrepresents the findings from this study.

With respect to the dosage chosen for StatinWISE;

We gave careful consideration to the dosage of statin to use in the trial and our final decision to use 20mg was based on two factors: i) 20mg atorvastatin is the most commonly prescribed dose in UK primary care (the study setting), and ii) the need for pragmatism and cost effectiveness. Using different dosages for different patients would have vastly increased costs and complicated logistics. Reviewer 1 raised concerns over our current design and its burden on participants and care-givers, and we believe that our design is as simple as possible. We recognise in our limitations on page 4 that this study will only assess the effects of a single statin at a single dose. We recognise that higher potency statins may have different effects, but this would be a different research question.

For absolute clarity, we have made this limitation clear on page 4.

b. Duration: The two month time period may seem at first glance to be adequate but it is not, and particularly not with a multiple on-off design. Two months is consistent with the duration used by the Phillips 2002 double-blind N-of-1 placebo-controlled dechallenge-rechallenge biopsy study – which it is, incidentally, inexcusable not to cite, for a study of such related design¹. However it should be noted that Dr. Phillips later found that it was not every patient for whom a two months period sufficed, based on patients in his statin myopathy clinic (personal communication). The two month time might also appear to comport with the two month time period in our analysis of percent recurrence of statin muscle symptoms with rechallenge². But our recurrence analyses, we considered those who had rechallenged for at least two months, and most who had been back on statin for at least two months had, naturally, been on statin for more than two months at the time they completed the survey. Moreover, our direct data clearly show that two months is not consistently sufficient. Direct data on patient-reported time to onset and recovery were provided in that study. Muscle symptoms do not necessarily follow the timecourse of statin effects on lipids, analogous to the fashion in which symptoms caused by hypoxemia outlast the duration of the hypoxemia and may be permanent. Mean time to first noted improvement following statin discontinuation was 8 weeks, after first statin usage leading to problem – muscle adverse effect or MAE – with maximal recovery taking a mean of 29 weeks (though, median times were shorter, at 2 and 8 weeks respectively). Confining analysis to those who provided timecourse data with first statin leading to MAE and statin rechallenge leading to MAE, mean time to recovery after first statin muscle AE was six weeks; but was twice as long – 12 weeks – following the rechallenge statin. Also, though the median time to recurrence with rechallenge was significantly shorter than with the first statin causing a MAE, some people nonetheless took months for recurrence (of muscle symptoms that also resolved with rechallenge, so were likely statin associated). If longer times are not permitted for onset and resolution, the study will ensure that symptoms, due to the statin will be difficult for patients to distinguish during the placebo and statin periods. I would advise that if the stated purpose is to assess whether problems like these are real, then data relevant to real-world timecourse be used to determine timecourse for the study, and to better assess study power based on the fraction expected to conform.

We gave careful consideration to the time periods of the trial. We are satisfied that our time periods will be sufficient given evidence from Cham et al (where symptoms improved in a median of two weeks following discontinuation and re-appeared two weeks after re-challenge),^[1] the STOMP trial^[2] and the PRIMO trial.^[3] These studies indicate that our time period would be sufficient to capture the majority of symptoms and have now cited these studies in the manuscript (page 10). We agree that we are unlikely to capture the symptoms for patients whose symptoms appear much later

than eight weeks, but since this is a N-of-1 trial, rather than a simpler rechallenge-dechallenge study, we may capture symptoms in these patients where statins are given for two treatment periods in a row. We will ensure that we raise this in the main trial publication.

c. Participants: Since the stated purpose is to clarify whether statins can cause muscle problems, or if problems reported by patients are legitimate, and the investigators purports to show an avenue physicians can use, participants should be restricted to those who i) reported experiencing muscle problems on statins at least twice (with dechallenge-rechallenge), and ii) did so on a clearly lower potency statin without at the time being on an interacting medication; and iii) in a less than 1.5 month time period (since participants begin answering questions before two months). The study is not meaningful if participants are exposed to a potentially lower potency agent than the one that caused the problem (due to the agent or concurrent exposures); or took a longer time; and it is important to ascertain the degree of need for a placebo-controlled blinded study, which will be quite costly. The need may be negated if unblinded dechallenge-rechallenge was relatively reliable.

Our research question asks whether participants' muscle symptoms occurring during statin use are caused by their statins. We are not trying to exclude the possibility that there exists a person for whom statins cause muscle symptoms. Our inclusion/exclusion criteria are as broad as possible to get participants representative of patients presenting at a UK general practice, who believe that their muscle symptoms are caused by statins.

d. Secondary outcomes: None are chosen to reflect the more commonly reported adverse effects. Sleep comes closest. Cognition, fatigue, neuropathy, and irritability are among the most common problems reported by patients, in a patient targeted database. Three have RCT studies supporting them, and neuropathy has both good quality dechallenge-rechallenge evidence with concurrent objective findings (alluded to elsewhere); and a large Danish national database study²⁴. Even though your study focuses on muscle symptoms, data collected on other adverse effects should be based on past experience so that the study does not underreport the potential impact of rechallenging on a statin. (Sleep can be retained.) Since the sample size required to show problems with, say, sleep was previously on the order of 1000 participants, any one of these problems is unlikely to be significantly increased; but you should look at the count of these particular symptoms reported, on statin vs placebo.

We are not aiming to ascertain the overall potential impact of statin side effects. Our study addresses a focused question regarding muscle symptoms. Participants will be asked about other symptoms every two months, however, our study is deliberately designed to provide high quality evidence on a single primary question.

e. Include narrative comments from patients about their experience. Allow them to comment on whether they feel the experience was more helpful or more harmful. (You only ask if it was helpful; for a study examining harm, it is critical to ask about harm.)

We thank the reviewer for this suggestion. Since we are some way from examining this issue, we may be able to make changes to this aspect of the study. The patient experience is of great importance to us, particularly if there is a possibility to extend our methodology to everyday practice. The most useful approach may be to ask about overall patient experience, rather than suggesting help or harm.

f. Since this is a study of persons who reported past muscle adverse effects, to see if they will report them again, CK assessment is de riger, at baseline (before any treatment or placebo, ideally twice), and at the conclusion of each treatment period. I would suggest cardiopulmonary function testing of the kind innovated by Dr. Phillips, particularly respiratory exchange ratio, as well²⁵.

Related to this, "Patients who have previously experienced rhabdomyolysis or myopathy will be excluded from the trial." Please clarify, how is myopathy and how is rhabdomyolysis defined for this purpose? How will you ascertain if these occur in patients?

We thank the reviewer for the suggestion. Patients are recruited directly by their personal GPs. This pragmatic trial has been approved to pre-screen patient medical records before they are invited to take part. Any patient where there is a clinical diagnosis of rhabdomyolysis or myopathy are automatically excluded. Additionally, when searching the register of patients, patients with a raised CK or ALT (5 times normal levels) anytime within the last 3 months are excluded. However, repeated CK and cardiopulmonary assessment are not part of our primary outcome of muscle symptoms, are not about patient safety, and would be a burden on both GP practice staff and patients, so we do not intend to measure these.

The GPs of all participating patients will be required to confirm whether or not the patients is eligible for the trial, including an assessment of previous rhabdomyolysis or myopathy. This has now been added to the manuscript (page 20).

For participants who develop muscle symptoms and consult their GP, a measurement of CK is currently recommended as part of usual clinical care.

g. Long term follow-up is imperative to include. I would also urge revision of the methods to accommodate a randomized pure placebo arm (or based on equivalence to this from Nordic Cochrane data, an arm randomized no-treatment, for the duration of the study), to assess whether subjecting patients who have had possible prior statin problems, to subsequent statin courses may accelerate subsequent physical decline. Several forms of evidence support the possibility of persistent physiological impairment even if there is clinical resolution, following statin adverse effects. These include: persistent biopsy findings of myopathy in a sizeable subset²⁶; and persistent cardiopulmonary abnormalities^{25, 27}. (These latter presumably also reflect predisposition – but the existence of predisposition is contingent on the possibility that there was in fact something to predispose toward. Additionally, related changes in normal patients on statins goes against this serving purely as predisposition²⁸.) Further factors pointing to persistence include shorter mean time to recurrence following rechallenge; and longer mean time to noting improvement following rechallenge's dechallenge (reviewed above); and partial persistence of symptoms in a sizeable subset (as above). Additionally, after statin neuropathy, which can arise with or independent of myopathy, there is "Clinical evidence of proximal and distal weakness and muscle fasciculations and persistent abnormalities of sensory conduction after recovery"²⁹.

The literature regarding long term side-effects of statins is far from certain and our study question focuses on whether the muscle symptoms experienced by participants during statin use are caused by their statins. Therefore, our focus is on symptoms arising during treatment periods. Also, long term follow-up is likely to be very underpowered for a study of 200 participants. Questions regarding long-term effects would be better investigated within large-scale parallel group RCTs.

Regarding the suggestion for a complete placebo arm, we felt that there would be ethical concerns regarding taking someone, who is amenable to rechallenge, off statins for a whole year, given the known benefits of statins.

h. Participants should, ethically, be informed that problems arising on statins do not consistently resolve fully, including muscle symptoms^{2, 8, 30, 31}. They should be advised that there have been a number of reports in which problems arising on statins resolved – clinically at least – after each (one or more) prior discontinuations, but then after the problem developed on the next statin, the problem has failed to recover, or recovery fully, on discontinuation. (This has also occurred for people who previously experienced no problem on a statin³¹.)

We believe that the evidence for long term effects of statins following discontinuation is far from certain and has not been demonstrated in blinded, randomised trials. The evidence cited by the reviewer is from unblinded studies, or with limited sample size.

Our participants are patients whose GP has recommended statin treatment and we rely on the GP to give the patient all information that would routinely be given to patients in this position who are recommended to take statins.

i. Ethics and Information to Participants:

i) Ethically, participants should be informed that there are no data to indicate the presence of net benefit of statins in persons who have experienced statin muscle problems. They should be advised that, even among those who have not had adverse effects (those who have had adverse effects may fare worse), the expected median survival benefit on statins may be quite low – according to one study, perhaps on the order of days, including for secondary prevention³². They should be informed of the range of reported adverse effects that can arise on statins.

The information provided to Participants necessarily must be based on the best available evidence which comes from systematic review of the evidence and the Summary of Product Characteristics (the information on which a drug is licence for use). Additionally, the responsibility for what is presented and how it is presented must remain the responsibility of (a) the patient group who help to draft the information and (2) the Ethics Committee and the Regulatory Agency (MHRA) who reviewed it. It cannot be based on individual opinions. Amendments will be made if new evidence becomes available or on the basis of feedback from participants.

ii) Participants should be informed that problems arising on statins do not consistently resolve fully, including for muscle symptoms^{2, 8, 30, 31}. They should be advised that there have been a number of reports in which problems arising on statins resolved – clinically at least – after each (one or more) prior discontinuations, but then after the problem developed on the next statin, the problem has failed to recover, or recovery fully, on discontinuation³¹.

As above

iii) It was stated that “Known serious adverse effects of statins (rhabdomyolysis and myopathy) are extremely rare.”

a) This is not a meaningful statement, because of effect modification. The rarity of problems depends entirely on characteristics of the patient (and treatment). Claims data, for instance, showed that the number needed to treat to have one case of rhabdomyolysis ranged from one in 22,727, to one in 484 for older patients with diabetes mellitus on statin plus fibrates, to one in 9.7 to 12.7 treated patients in a special treatment setting³³ – that setting involving the highest frequency will not occur now, but there will be other specialized subsets that will have markedly elevated risk for serious problems due to a confluence of risk factors, including some we may not have identified. {Similarly our RCT data affirmed effect modification – again showing that subsets – again, older age coupled with metabolic syndrome factors – had markedly elevated risk of glucose elevation due to statin (i.e. compared to placebo); and while older age plus female sex (each a known risk factor for statin problems) markedly increased risk of muscle weakness – replicated on each simvastatin vs placebo and pravastatin vs placebo, so not plausibly attributable to a chance finding due to multiple looks.

b) Second, it is vital to be clear to patients that by the definition employed, ongoing suffering from pain, for instance, is not counted as “serious.”

Whether or not effect modification exists, our statement about the overall prevalence of rhabdomyolysis and myopathy remains unarguable.

Importantly, in our study, all patients will have been previously exposed to statins and any with previous rhabdomyolysis or myopathy will be excluded. Any patient developing rhabdomyolysis or myopathy during the trial will be withdrawn. The patients included in our study will therefore be at extremely low risk of these outcomes. Our participant information sheet clearly explains that we are inviting patients to be in the study because they are experiencing or have experienced muscle symptoms, and want to help them ascertain whether they appear to be related to their statin use.

j. Exclusions include “Patient has persistent, generalised, unexplained muscle pain (whether associated or not with statin use) and blood creatine kinase (CK) levels greater than 5 times the upper limit of normal”

a) As elsewhere in this review, patients may have serious statin-induced myopathy without CK elevations.

b) Additionally, some prior “muscle” problems may be due to neuropathy^{24, 29, 34}, not associated with CK elevation. Risk of these problems may particularly accrue with time of statin use^{35, 36}, and though many or most recover partially³⁷, many do not recover fully³⁸; moreover, even where there is clinical recovery, there may be persistent pathological injury²⁹; and one prospective study found development of pathological changes, even in absence of clinical ones³⁹.

c) These patients – at potentially heightened risk of statin adverse effects (based on prior possible adverse effect) – should be provided with this information prior to accepting rechallenge on statins. The use of statins in this study is entirely within their drug licence with all cautions and contraindications applied as decided by the UK drug regulator, the MHRA.

Analyses:

Comment: Sample size calculations are problematic. The study is powered based on the notion that a 1cm VAS difference is relevant, and that that must be the average effect. But the whole premise of the study is that not everyone in whom muscle problems developed on statins, developed these due to statins, ANY significant difference will show that some patients developed problems causally due to statins. Use of a 1cm threshold, after enrolling a sample of uncertain causality, appears to reflect a determination not to show an effect. And the sample size should be inflated to consider your perspective that not all prior reports need have been causal.

“Using simulation, we estimated a sample size...” – with what inputs? On what basis?

Response: We have described the power calculations in more detail than is commonly employed, due to the level of complexity in these calculations. We do not feel that further description of the simulation design would be illuminating to the reader. However, one aspect of the simulations is important to stress – we considered the possibility of heterogeneity in treatment effects. This is the main driver behind inflating the sample size from 107 to 200.

Any power calculation requires specification of a minimum clinically significant difference, which is always expressed as an average effect. Our choice of 1cm is consistent with a previous pilot study addressing the same question. Whatever our overall findings, we will publish our overall point estimates, 95% confidence intervals, and estimated treatment heterogeneity (as a variance component). Thus, if there is an effect of statins on muscle symptoms, but on average this is less than 1cm, these estimates will provide an indication of the possible magnitude of such an effect. Our estimates can be combined with estimates from similar studies in the future, in order to provide power to detect smaller effects.

Comment: Be sure to analyze dropouts as a critical outcome. Participants who drop in the first period – or in the second period prior to VAS – are not considered.

Participants who dropout prior to providing a VAS measurement on both statin and placebo are not included in the primary analysis, because their data holds no statistical information regarding that comparison. In line with current recommendations for best practice for handling missing data in trials, we will undertake sensitivity analyses which assess the robustness of our results to a range of missing data mechanisms. One of these will be a multilevel model, similar to the primary analysis, additionally incorporating subjects who provided outcome data only under one treatment condition.

Response: We will report the number of dropouts by period and treatment allocation, as per the Consort extension for N-of-1 trials.

Comments: Please include simple analysis to show initial effects:

- Change in mean score from no treatment onto statin, and from no treatment onto placebo
- Change in mean score from first treatment phase to second: placebo to statin; and from statin to placebo
- Then mean of change in mean, for each statin-to-placebo, and each placebo-to-statin
- Dropout in first phase if placed on statin; dropout in first phase if placed on placebo
- Dropout in first statin phase, whichever time; compared to dropout in first placebo phase
- Inquire about cause of dropout and analyze

We will include graphical descriptions of the trajectory of participants through the course of the trial – in terms of their VAS scores – as part of our descriptive analysis. As stated in response to the previous comment, dropout will be reported and described as per the Consort statement.

Analyze the percent who develop symptoms overall in addition to the percent that attribute to statins on statin vs placebo. These can and should be a simple analysis.

Our trial statistician disagrees that this is a simple analysis! However, this will indeed be undertaken. As described in our manuscript, this will be performed via a mixed effect logistic regression model, to account for within-participant clustering of symptoms over time. We will also report percentages with symptoms by period and treatment allocation.

Separately analyze the first and second phases as longer use may lead to longer recovery time and more difficulty distinguishing statin from placebo

This analysis will not form part of our primary analysis, but we will explore such patterns in sensitivity analyses. These will be reported and interpreted as sensitivity analyses.

Please clarify what this means. “We will investigate whether the excess muscle symptoms, if any, experienced during treatment periods with the statin appears to be concentrated in multiple sites.” “concentrated” and “multiple” are a confusing pair of words.

Response: We thank the referee for this useful request for clarification. We have now amended the wording to state:

“We will investigate whether the excess muscle symptoms, if any, experienced during treatment periods with the statin are at a single body site versus multiple sites; and whether symptoms are more common at any specific body site.”

The meaning of the statement has not changed.

Discussion

Limitations: Self-selection was partially addressed. However, potential for extreme self-selection should be more fully acknowledged. Patients who have strong cause to believe their problem was due to a statin will be justifiably disinclined to rechallenge. Those whose problems, legitimately due to statins, were persistent may be justifiably disinclined to participate⁸. Participants may (indeed should) primarily be those whose problem was equivocally present, and/or who aren't persuaded the problem was necessarily due to the statin. Thus it will assess the subgroup in whom a causal relationship is least likely, and for whom reexposure to statins may least likely produce problems. Generalizations will be made from these findings to patients whose problems are more likely related to statins, and more serious – in whom the peril of rechallenge will be potentially far greater. This limitation should be fully acknowledged; it should be made clear that findings for these patients cannot be thought to generalize to other patients who reported muscle problems on statin, and did not elect to participate. As the reviewer states, we have acknowledged the possibility of self selection and are unable to make this point more clearly within the constraints of a protocol paper. This is a key point that we intend to return to when the trial results are published.

Citations: The citations are weighted strongly toward articles that disclaim most statin adverse effects – and the present article appears to accept such characterizations. These include citation of the 2016 Lancet “review” by Collins et al⁴⁰, on which the corresponding author for this paper was a coauthor; and the article by Finegold⁴¹ et al on which another of the investigators participated (none of the considerations exposed in a rebuttal to that article are reflected¹⁵). A further citation, with first author Armitage⁴² (an investigator on this study), states that even serious muscle adverse effects/ rhabdomyolysis usually reverse fully. Based on what? This does not comport with our experience. As Dr. Phillips noted, persistence of weakness was more common in rhabdomyolysis than in non-CK elevating statin muscle problems; but even in the latter, in our published data on non-rhabdomyolysis statin muscle problems, a minority reported that recovery was complete². As Dr. Phillips, with his statin myopathy clinic, noted, the fact that a patient stopped complaining (or showing up) does not imply their recovery was complete³⁰. And Dr. Vladutiu noted, in her study examining genetic predisposition to statin muscle problems, “Variable persistent symptoms occurred in 68% of patients despite cessation of therapy”⁸. Other citations such as those of Barsky and Zhang fit the slant of the citations.

There is currently no meaningful reflection (or citation) of relevant materials providing contrary data and inference, such as those documenting mechanism (and concurrent findings like fatty acid oxidation and respiratory exchange ratio changes), disability, and dose-response – or symptom persistence^{1, 6-9, 25, 30, 43}; supporting the presence of adverse effects (including by documenting risk factors for them), or challenging the perspectives of the cited documents^{9, 14, 15}. To allow an appearance of impartiality on the issue, or to allow the perception that this is an honest effort to examine the purported topic of this study, the authors need to make an aggressive effort to portray both knowledge of evidence on the other side, and an actual openness to the possibility that statins may causally produce problems they purport to be studying. Ideally they should incorporate individuals in their investigator group who are experts on these statin muscle problems (such as Dr. Paul Phillips, Dr. Georgirene Vladutiu, Dr. Helmut Sinzinger, and/or Dr. P. Thompson), if any are ethically comfortable with participating. This would assist in development of a design more advantageous to detecting problems if they occur; and to negate the appearance that the study’s purpose, based on past articles by the authors, and the selection of citations, may be to not find problems.

As stated above, we have now added citations to reflect some of the opinions regarding resolution of symptoms. However, much of the literature cited by the reviewer originates from unblinded studies, which are more prone to bias than the studies that we originally cited.

Statements needing revision

“Clinicians faced with patients presenting with muscle symptoms during statin use are encouraged to measure the blood creatine kinase, which is substantially elevated in the rare cases of statin associated myopathy, but in the vast majority of patients the CK level will be normal, ruling out myopathy.” {emphasis mine}

- This statement suggests ignorance of the evidence (or distortion). Data by Phillips^{1, 30} (and others) clearly show that non-CK elevating muscle symptoms on statin can absolutely represent myopathy. The reconstruing of myopathy, in some publications, to require high CK elevation (with an arbitrary cutoff) to report myopathy on statins, seems clearly designed to reduce acknowledgment of problems. We respect the opinion of the reviewer but do not feel that we need to address this point in the manuscript.

“Strengths

“This trial will determine whether statins cause muscle symptoms during statin use in these participants, allowing clinician and participant to make an informed decision about continued use.”

- It will only determine whether atorvastatin at 20mg given for 2 months (with 2 months allowance for recovery), does so, in patients who may only have had problems on higher potency agents.

One of the stated limitations on page 4 is that our trial will only assess the effects of a single statin at a single dose.

“This trial addresses some of the limitations of previous statin side effects literature.”

- Toward what purpose? Which limitations? There is already a blinded placebo-controlled dechallenge-rechallenge study, that goes farther than this by adding biopsy assessment; this and other data eliminate doubt about the existence of statin muscle adverse effects. This study has different significant limitations, which should be better acknowledged.

As above, we are aware of the limitations of StatinWISE and have acknowledged the major limitations on page 4.

“In these cases there is currently no other diagnostic tool allowing clinicians to empirically evaluate whether symptoms reported by an individual statin-user are caused by the statin itself or by the ‘nocebo’ effect.”

- Since the nocebo effect has less evidence than causal statin muscle effects (reviewed below), this statement needs to be revised. The necessity and burden of a placebo dechallenge is likely to be seldom necessary, particularly given the evidence of quite modest expected benefits of statins³². The first approach is to reevaluate how compelling is the need for the drug. Then, if necessary, a trial of dechallenge-rechallenge could be attempted.

We respect the opinion of the reviewer but do not feel that we need to address this point in the manuscript.

“Randomised controlled blinded trials (RCTs) have found no evidence of an effect on these symptoms.^{12}”

- As repeatedly reported, because of effect modification, average effects from a randomized trial that fail to show significant harm to an outcome in the treatment group, have little bearing on whether the drug can cause that harm in a vulnerable subgroup. This has been shown across many treatments – bisphosphonates can cause fracture; anxiolytics can cause anxiety; antidepressants can cause suicide; antihypertensives can “paradoxically” raise blood pressure, as can salt restriction. Antioxidants can almost always have prooxidant effects, and this likely underlies statins’ bidirectional effects on many outcomes⁹.

- Statins may reduce risk of muscle pain in some, by improving energy and oxidative stress, just as it contributes to them, impairing energy and worsening oxidative stress in others⁹. That some may benefit on the drug, does not reduce the impact or drug contribution for those who are harmed. Analogously, statins reduced diabetes in the WOSCOPS trial⁴⁴ but increase it in studies with high potency or participants with risk factors for statin problems – e.g older age or metabolic syndrome factors (which are also risk factors for diabetes). Moreover, in some studies (like JUPITER), neutral average effects on glucose accompany increases in incident diabetes, comporting with bidirectional effects. The only means by which statin muscle problems can be challenged, is via methodological failure – failure to recognize and understand the issue of “effect modification,” which causes bidirectional effects with many drugs and exposures (and particularly ones that have antioxidant effects, which typically provides prospects for prooxidant effects)⁴⁵.

The reviewer clearly believes that there are a set of (currently unidentifiable) individuals for whom statins might cause muscle pain. One of the benefits of the N-of-1 design is that we can estimate individualized effects, so part of the primary analysis will comprise reporting the variability of treatment effect estimates across participants in the trial. If we find such variability, we can subsequently pool our data with other similar studies to try and identify characteristics of such patients.

- Additionally, that statement is untrue. E.g.:

o Although this has been published only as an abstract, it was shown clearly that by coupling known risk factors to identify a high risk group within a randomized trial (because, again, of recognition of the existence of effect modification), such as female sex and older age, muscle weakness was significantly greater on statin than on placebo. This was separately significant for simvastatin vs placebo and for pravastatin vs placebo, so is not plausibly attributable to chance⁴⁶.

This was from a post-hoc subgroup analysis. Multiple testing cannot be ruled out and this is not yet a published paper.

o A significant increase in fatigue was also found in the full sample for that RCT⁴⁷. The article reports on a composite measure (fatigue and exertional fatigue) because this was demanded by reviewers; but the original submission showed significance separately for the two fatigue-related measures, so that concerns based on composite measures do not apply.

This study investigates fatigue. Our research question is about muscle symptoms.

o And one of the participants on this WISE effort was a coauthor on the “Search” trial. What of the evidence procured on what was termed “incipient myopathy”? (really, two-organ, combined muscle and liver adverse effects – but with muscle problems short of rhabdomyolysis).

Our trial investigates muscle symptoms that fall short of myopathy. Any patient with myopathy would be excluded from the StatinWISE study.

o There is also the Silver 2006 meta-analysis of RCTs. “Statin therapy increased the risk of any AE by 39% (OR = 1.4; 95% CI, 1.09-1.80; P = 0.008.... Nonurgent AEs such as myalgia and liver function elevations were responsible for approximately two thirds of AEs reported in trials.”

This meta-analysis does not report on myalgia, or other specific adverse events, comparing placebo and statin.

Comment: Rhabdomyolysis has also generally not shown up as a statin AE in RCTs and meta-analyses of them:

Why are other forms of evidence accepted for rhabdomyolysis, but not muscle problems short of this?

Response: Rhabdomyolysis is an extremely rare outcome. Therefore in trials designed and adequately powered to detect differences in cardiovascular disease outcomes between arms, we would not expect to see an effect on rhabdomyolysis. However, reports of muscle symptoms while on statins are much more frequent. One could argue that this much higher prevalence of muscle symptom reports suggests that we should see an effect on muscle symptoms in the larger statin trials, if there is one. However, our argument is that the trials have not shown an effect but patients continue to be concerned about muscle symptoms and stop their statins accordingly, hence the uncertainty of effects. Our trial aims to demonstrate whether or not statins are the cause of muscle symptoms, among patients that take part.

Comment: “A recent review of the evidence from randomised trials and observational studies suggested that symptomatic adverse events may be misattributed to statins.^{9}”

- The authors elected to ignore the issue of effect modification, though the importance of this methodological consideration in relation to statin AEs has been raised repeatedly^{9, 14, 15}, including in a published response to a similar claim¹⁵.

Response: We have responded to the reviewer’s comments about effect modification above. We do not feel that this statement needs to be revised as it reflects the findings of the cited study.

Comment: “Uncertainty about the association between muscle symptoms and statins persists due to limitations of observational studies and trials.”

- Any uncertainty necessitates either inadequate knowledge of the literature, or inadequate methodological aptitude.

Response: We respect the opinion of the reviewer but do not feel that we need to address this point in the manuscript.

“If the CK and ALT are within the normal range and the participant remains eligible they can be offered the following options by the GP:”

Comment: What exactly is the plan if they are abnormal? Is there no concern that these may be very high before the participant cites symptoms, in someone selected for potentially being at risk?

Response: Patients CK and ALT levels will be measured prior to trial entry and those whose levels are abnormal will not be eligible for the trial. As in everyday practice, when the patient complains of serious muscle pains, they will be offered a blood test to measure CK and ALT, and if raised then the patient will be withdrawn from statins. Our trial mirrors standard care.

Comment: “A key secondary outcome is pain for which the participant answers “yes” or “don’t know” to the question “Do you think these muscle symptoms are related to your study medication?” asked at the end of each treatment period. People answering “No” to this question would have their VAS scores set to zero for that treatment period for this specific secondary outcome.”

- Please be more clear about whether this “zero” just refers to “no” (don’t think it is related), or refers to the outcome of pain?

Response: This analysis will compare VAS muscle symptom scores between statin and placebo periods, but setting patients’ VAS scores to zero for that period if they say they do not believe their symptoms (for that period) to be related to their study medication (which may be statin, or may be placebo). This is a secondary analysis, intended to remove symptoms that the participant attributes to an external event (e.g. running a marathon the previous day).

Comment: What is the rationale for looking at genetic predictors of statin effects if you are not persuaded these exist?

Response: The best reason for doing a study is when there is uncertainty. Genetic factors are linked to statin-related myopathy and more research is needed in this area to understand the further effects that genetics can play in the link between statins and side effects. Given the nature of the trial and the need to draw blood to ascertain eligibility, it is an opportune time to contribute to the ongoing research into this area.

Comment: “A major limitation of observational studies is a lack of blinding; patients taking a medication expect to experience adverse effects^{13} and therefore reporting of symptoms in statin users may be higher than in a comparable population not on statins. This phenomenon, the ‘nocebo’ effect, can lead to bias in unblinded studies.”

- This needs to be revised based on the actual evidence regarding placebo-nocebo effects. Unlike statin adverse effects (wherein effect modification renders average effects of RCTs irrelevant to subsets), the whole premise of placebo effects is the idea that on a placebo, randomized against an active drug, expectation produces average differences in reporting relative to if there had been no treatment. This is empirically testable, and the determination should rely on controlled trials. Controlled studies, reviewed by the Nordic Cochrane Center, in fact show little evidence for any placebo effects (Nordic Cochrane Center data)^{23, 48}: In trials that include a randomized “no treatment” arm together with a randomized placebo arm, evidence fails to validate net placebo effects except for analgesic effects (which are shown in a separate literature to be mediated by release of endogenous opiates, and blocked by naloxone – i.e. their mechanism that does not generalize to other treatments and outcomes). What people have attributed to placebo/ nocebo effects derive from other considerations (including, in some cases, active effects of the placebo⁴⁹⁻⁵¹.)

Response: We disagree with the opinion of the reviewer regarding the nocebo effect. However, we feel that there is good evidence for a nocebo effect in the context of statin side effects. First, in the Odyssey ALTERNATIVE trial,[4] statin 'intolerant' patients initially underwent a double blinded four week phase where they received placebo. During this time, 7% dropped out due to myalgias. In the main phase of this three armed trial (alirocumab vs ezetimibe vs atorvastatin), rates of adverse events were the same across all groups at roughly 80%, but dropped to 55% among alicumab when unblinded.[4] Second, a recent paper in the Lancet from the ASCOT trial showed that patients were more likely to complain about muscle symptoms once they were told that they were on a statin.[5] We have now added references to these studies in the manuscript to support our statement.

Comment: To summarize, this has potential to be a useful study, and there exists the prospect of an acceptable manuscript in relation to this. For this to be the case, the manuscript must better reflect and acknowledge existing evidence that documents existence of statin muscle adverse effects, including effects short of rhabdomyolysis. It must characterize its purpose differently; it is not ethical to expose patients to a drug, who may have already responded to that drug adversely, for the purpose of showing something that is already known. It must revise the design, capitalizing on existing information to better guide dose and duration, in order to have ability to assess reproducibility of adverse muscle effects arising on statins; it must better acknowledge the uncertainties in risks and benefits that may redound to the patient, both in the article and to prospective participants themselves. It must more openly characterize its limitations.

Response: Once again, we thank the reviewer for her careful consideration of our manuscript. We believe that our carefully designed study can answer our intended question regarding the muscle-related side effects of statins. In the manuscript, we acknowledge many of the limitations raised by the reviewer and have added further clarification and response where we feel this adds to the paper.

References

1. CHAM, S., EVANS, M. A., DENENBERG, J. O. & GOLOMB, B. A. 2010. Statin-associated muscle-related adverse effects: a case series of 354 patients. *Pharmacotherapy*, 30, 541-53.
2. PARKER, B. A., CAPIZZI, J. A., GRIMALDI, A. S., CLARKSON, P. M., COLE, S. M., KEADLE, J., CHIPKIN, S., PESCATELLO, L. S., SIMPSON, K., WHITE, C. M. & THOMPSON, P. D. 2013. Effect of statins on skeletal muscle function. *Circulation*, 127, 96-103.
3. BRUCKERT, E., HAYEM, G., DEJAGER, S., YAU, C. & BEGAUD, B. 2005. Mild to moderate muscular symptoms with high-dosage statin therapy in hyperlipidemic patients--the PRIMO study. *Cardiovasc Drugs Ther*, 19, 403-14.
4. MORIARTY, P. M. 2014. ODYSSEY ALTERNATIVE demonstrates complexity of statin-intolerant patients. *AHA* 2014.
5. GUPTA, A., THOMPSON, D., WHITEHOUSE, A., COLLIER, T., DAHLOF, B., POULTER, N., COLLINS, R. & SEVER, P. 2017. Adverse events associated with unblinded, but not with blinded, statin therapy in the Anglo-Scandinavian Cardiac Outcomes Trial-Lipid-Lowering Arm (ASCOT-LLA): a randomised double-blind placebo-controlled trial and its non-randomised non-blind extension phase. *Lancet*.

References

1. Phillips PS, Haas RH, Bannykh S, et al. Statin-associated myopathy with normal creatine kinase levels. *Ann Intern Med* 2002;137:581-5.
2. Cham S, Evans MA, Denenberg JO, Golomb BA. Statin-associated muscle-related adverse effects: a case series of 354 patients. *Pharmacotherapy* 2010;30:541-53.
3. Hoffman KB, Kraus C, Dimbil M, Golomb BA. A Survey of the FDA's AERS Database Regarding Muscle and Tendon Adverse Events Linked to the Statin Drug Class. *PLoS One* 2012;7:e42866.

4. Dale KM, Coleman CI, Henyan NN, Kluger J, White CM. Does More Aggressive Statin Therapy Increase Muscle and Liver Risk? In: 55th Annual Scientific Sessions of the American College of Cardiology; 2006 Mar 11-14; Atlanta, GA; 2006.
5. Link E, Parish S, Armitage J, et al. SLCO1B1 variants and statin-induced myopathy--a genomewide study. *N Engl J Med* 2008;359:789-99.
6. Sinzinger H, Lupattelli G, Chehne F, Oguogho A, Furberg CD. Isoprostane 8-epi-PGF2alpha is frequently increased in patients with muscle pain and/or CK-elevation after HMG-Co-enzyme-A-reductase inhibitor therapy. *J Clin Pharm Ther* 2001;26:303-10.
7. Phillips PS, Ciaraldi TP, Kim DL, Verity MA, Wolfson T, Henry RR. Myotoxic reactions to lipid-lowering therapy are associated with altered oxidation of fatty acids. *Endocrine* 2009;35:38-46.
8. Vladutiu GD, Simmons Z, Isackson PJ, et al. Genetic risk factors associated with lipid-lowering drug-induced myopathies. *Muscle Nerve* 2006;34:153-62.
9. Golomb BA, Evans MA. Statin adverse effects : a review of the literature and evidence for a mitochondrial mechanism. *Am J Cardiovasc Drugs* 2008;8:373-418.
10. Mammen AL, Chung T, Christopher-Stine L, et al. Autoantibodies against 3-hydroxy-3-methylglutarylcoenzyme A reductase in patients with statin-associated autoimmune myopathy. *Arthritis Rheum* 2011;63:713-21.
11. Grable-Esposito P, Katzberg HD, Greenberg SA, Srinivasan J, Katz J, Amato AA. Immune-mediated necrotizing myopathy associated with statins. *Muscle Nerve* 2010;41:185-90.
12. Liang C, Needham M. Necrotizing autoimmune myopathy. *Curr Opin Rheumatol* 2011;23:612-9.
13. Barbiroli B, Frassinetti C, Martinelli P, et al. Coenzyme Q10 improves mitochondrial respiration in patients with mitochondrial cytopathies. An in vivo study on brain and skeletal muscle by phosphorous magnetic resonance spectroscopy. *Cell Mol Biol (Noisy-le-grand)* 1997;43:741-9.
14. Golomb BA. The importance of monitoring adverse effects in statin, and other, clinical trials. *Clin Invest* 2013;3:913-6.
15. Golomb BA. Misinterpretation of trial evidence on statin adverse effects may harm patients. *Eur J Prev Cardiol* 2015;22:492-3.
16. Turner EH, Knoepflmacher D, Shapley L. Publication bias in antipsychotic trials: an analysis of efficacy comparing the published literature to the US Food and Drug Administration database. *PLoS Med* 2012;9:e1001189.
17. Stelfox HT, Chua G, O'Rourke K, Detsky AS. Conflict of interest in the debate over calcium-channel antagonists. *N Engl J Med* 1998;338:101-6.
18. Bero L, Oostvogel F, Bacchetti P, Lee K. Factors Associated with Findings of Published Trials of Drug-Drug Comparisons: Why Some Statins Appear More Efficacious than Others. *PLoS Med* 2007;4:e184.
19. Barnes DE, Bero LA. Why review articles on the health effects of passive smoking reach different conclusions. *Jama* 1998;279:1566-70.
20. Harris G. Caustic Government Report Deals Blow to Diabetes Drug. *New York Times* 2010;July 9, 2010.
21. Did GSK trial data mask Paxil suicide risk? *The New Scientist* 2008; 8 February 2008 12.
22. Smith R. Conflicts of interest: how money clouds objectivity. *J R Soc Med* 2006;99:292-7.
23. Hrobjartsson A, Gotzsche PC. Placebo treatment versus no treatment. *Cochrane Database Syst Rev* 2003;CD003974.
24. Gaist D, Jeppesen M, Andersen LA, Garcia Rodriguez J, Hallas J, Sindrup SH. Statins and risk of polyneuropathy: A case-control study. *Neurology* 2002;58:1333-7.
25. Phillips PS, Phillips CT, Sullivan MJ, Naviaux RK, Haas RH. Statin myotoxicity is associated with changes in the cardiopulmonary function. *Atherosclerosis* 2004;177:183-8.
26. Mohaupt MG, Karas RH, Babychuk EB, et al. Association between statin-associated myopathy and skeletal muscle damage. *CMAJ* 2009;181:E11-8.
27. Phillips CT, Gray NL, Puhek LM, McDonald FG, Sullivan MJ, Phillips PS. Cardiopulmonary Function Is Abnormal After Statin-Induced Rhabdomyolysis and Myositis [abstract]. *J Am Coll Cardiol* 2004;43 (suppl A):16a.

28. Phillips CT, Gray NL, Puhek LM, McDonald FG, Sullivan MJ, Phillips PS. Basal Respiratory Exchange Ratio Is Altered With Statin Use in Normals [abstract]. *J Am Coll Cardiol* 2004;43 (suppl A):233a.
29. Phan T, McLeod JG, Pollard JD, Peiris O, Rohan A, Halpern JP. Peripheral neuropathy associated with simvastatin. *Journal of Neurology, Neurosurgery and Psychiatry* 1995;58:625-8.
30. Phillips PS, Haas RH. Observations from a statin myopathy clinic. *Arch Intern Med* 2006;166:1232-3.
31. Koslik HJ, Meskimen AH, Golomb BA. Physicians' Experiences as Patients with Statin Side Effects: A Case Series. *Drug Safety - Case Reports* 2017;4:3.
32. Kristensen M, Christensen P, Hallas J. The effect of statins on average survival in randomised trials, an analysis of endpoint postponement. *BMJ Open* 2015;5:e007118.
33. Graham DJ, Staffa JA, Shatin D, et al. Incidence of hospitalized rhabdomyolysis in patients treated with lipid-lowering drugs. *Jama* 2004;292:2585-90.
34. Camargos EF, Oliveira Lde F, Boaventura Tde D. My legs are getting old: simvastatin-induced polyneuropathy. *BMJ Case Rep* 2011;2011.
35. Jeppesen U, Gaist D, Smith T, Sindrup SH. Statins and peripheral neuropathy. *Eur J Clin Pharmacol* 1999;54:835-8.
36. de Langen JJ, van Puijenbroek EP. HMG-CoA-reductase inhibitors and neuropathy: reports to the Netherlands Pharmacovigilance Centre. *Neth J Med* 2006;64:334-8.
37. Backes JM, Howard PA. Association of HMG-CoA reductase inhibitors with neuropathy. *Ann Pharmacother* 2003;37:274-8.
38. Adverse Drug Reactions Advisory Committee (ADRAC). Statins and peripheral neuropathy. *Australian Adverse Drug Reactions Bulletin* 2005;24:6.
39. Otruba P, Kanovsky P, Hlustik P. Treatment with statins and involvement of the peripheral nervous system: results of a prospective clinical and neurophysiological follow-up. *Biomed Pap Med Fac Univ Palacky Olomouc Czech Repub* 2007;151:307-10.
40. Collins R, Emberson J, Armitage J, et al. Interpretation of the evidence for the efficacy and safety of statin therapy. *TheLancetcom* 2016; Published Online September 8, 2016 [http://dx.doi.org/10.1016/S0140-6736\(16\)31357-5](http://dx.doi.org/10.1016/S0140-6736(16)31357-5).
41. Finegold JA, Manisty CH, Goldacre B, Barron AJ, Francis DP. What proportion of symptomatic side effects in patients taking statins are genuinely caused by the drug? Systematic review of randomized placebo-controlled trials to aid individual patient choice. *Eur J Prev Cardiol* 2014;21:464-74.
42. Armitage J. The safety of statins in clinical practice. *Lancet* 2007.
43. Vladutiu GD. Genetic predisposition to statin myopathy. *Curr Opin Rheumatol* 2008;20:648-55.
44. Freeman DJ, Norrie J, Sattar N, et al. Pravastatin and the development of diabetes mellitus: evidence for a protective treatment effect in the West of Scotland Coronary Prevention Study. *Circulation* 2001;103:357-62.
45. Golomb B. Effect modification. In: Edge.org, ed. *What Scientific Term or Concept Ought to Be More Widely Known*; 2017.
46. Golomb BA, Koperski S. Who becomes weak on statins? Effect modification exposed in a RCT by risk factor compounding. *Circulation* 2013;127:AP072.
47. Golomb BA, Evans MA, Dimsdale JE, White HL. Effects of statins on energy and fatigue with exertion: results from a randomized controlled trial. *Arch Intern Med* 2012;172:1180-2.
48. Hrobjartsson A, Gotzsche PC. [What is the effect of placebo interventions? A systematic review of randomized clinical trials with placebo treated and untreated patients]. *Ugeskr Laeger* 2002;164:329-33.
49. Golomb B. When are medication side effects due to the nocebo phenomenon? *Jama* 2002;287:2502-3; discussion 3-4.
50. Golomb BA. Paradox of placebo effect. *Nature* 1995;375:530.
51. Golomb BA. Are placebos inert or powerful? Vice versa. *Clinical Investigation* 2011;1:1471-3.

VERSION 2 – REVIEW

REVIEWER	Gabriel Chodick Maccabi Healthcare Services, Israel No competing interests.
REVIEW RETURNED	22-Jul-2017

GENERAL COMMENTS	The authors have adequately addressed my comments.
--

REVIEWER	Johann Auer, MD Department of Cardiology, St. Josef Hospital Braunau, Austria none
REVIEW RETURNED	15-Jul-2017

GENERAL COMMENTS	This study focus on muscle symptoms. However, statins are associated with other skeletal-related symptoms beyond muscles. Assessment of such side effect would not increase burden on participants and GPs that much. In contrast, assessment of such symptoms would give further insights into this issue. Assessment of LDL-C is not an unusual procedure during treatment with a cholesterol-lowering drug. Most patients are aware of LDL-C at baseline and of changes of LDL levels during treatment. Thus, blinding may be compromised by (unscheduled) LDL-C. measurements during the study. This important issue remains and should be addressed.
--

REVIEWER	Beatrice A. Golomb, PhD University of California, San Diego, La Jolla, CA, United States I am the executor of an estate (and will be among the beneficiaries) that has some stock in statin manufacturing pharmaceutical companies. I have been advised by the estate lawyer to make no changes to the stock portfolio until the time of distribution.
REVIEW RETURNED	04-Aug-2017

GENERAL COMMENTS	Broad statements: I. Comment on reply opening: The authors begin their reply not by a point by point response to the substance of comments, but by a sweeping false and offensive characterization of the reviewer – who gave considerable time, and shared strong expertise on statin effects and study methodology, for no compensation, to provide information and suggestions of relevance to improving the authors' manuscript and the underlying study. While maligning the referee so as not to be responsible for fully addressing the content of the referee comments may be an effective rhetorical tool, the approach also constitutes an acknowledged logical fallacy – colloquially, “poisoning the well.” It is also highly unprofessional. The points I made in no instance reflect “personal preference” as the authors assert, except preference for high quality evidence and inference, and patient protection – each of which should be the authors' preferences as well.
---

Nor do the comments reflect opinion other than that guided by sound inference from a more in depth knowledge of the evidence on this topic than most will have (including, almost certainly, those who approved the study proposal, as the latter are not selected for their expertise on this specific issue). Moreover in virtually all instances a literature foundation for statements was cited. (If there is any point which the reviewers feel to be lacking in cited evidence, I will be happy to share the undergirding evidence if asked.)

The authors comments about evidence suggest a belief that existing RCTs are free from bias and inherently and always provide high quality evidence to address every question, moreso than other study designs – and irrespective of what those RCTs were designed to address, how they were designed, what selection characteristics they had, what outcomes they assessed and how, and who the funders were. This is a simplistic and fallacious perspective, that fails to grasp the impact of key factors such as self-selection and selection bias that can seriously compromise external validity; and the well-documented and often profound impact of drug company conflict of interest on reported study results and conclusions. These are not issues of my “personal preference”; these are basic considerations of scientific methodology powerfully buttressed by empirical evidence.

Apropos of this, if my inferences depart from those of others, this cannot be presumed to reflect unfavorably on me. The statin literature (and guidelines, as journalists have noted¹⁻³) is driven heavily by parties and studies with industry conflict of interest^{4, 5}. Data (not “personal preference” or “opinion”) show that such industry conflicts of interest are tied to not small, but often radical shifts in stated results⁶⁻⁸, and even moreso conclusions⁶⁻⁹ (as well as stated opinions¹⁰). {Of note, the groundbreaking experiments of pioneering social psychologist Solomon Asch show that once there are individuals who have willfully distorted results and conclusions, many others will believe the distorted conclusions even when this requires disregarding the obvious evidence before their eyes.}

II. I am sure it is frustrating to receive feedback that asks for additional effort for a study that has already been approved.

III. Please understand that my comments are intended to protect against study flaws including and especially those that are expected to bias results to the null and may lead to flawed inferences that may place future patients in needless peril. Better the authors’ frustration now, then harm to patients later. For this study more than for most, the importance of sound design cannot be overstated.

A) The study asks that participants place themselves at possible risk, and this should only be done with the highest quality design, and the greatest protection against type II error.

B) If the inferences drawn from the study are in error, in the direction of fostering false dismissal of patients’ adverse effect experiences, the study may lead to subjecting many future patients (potentially including these specific ones) to needless suffering. Because of this it is vital that the authors take measures to ensure the design does not foster false negative results. This study has potential to make an important contribution; I would like for it to do so.

IV. There are high quality data supporting all of the key points I had made – such as for dose-dependence of statin adverse effects, (self)-selection bias, effect modification, and the influence of conflict of interest on evidence.

Even had these not been separately characterized in the case of statin adverse effects, they are known considerations in relation to study design and interpretation generally. That is, addressing limitations based upon these is not an expression of “personal preference” but appropriate recognition of important methodological considerations. (Understandably, the authors’ “personal preference” may be to discount such evidence selectively where it is disadvantageous.)

V. As I see it, the options are:

A. The authors can put in effort now, to make this a better quality study – even a good quality study with improved prospects to address its supposed goals. To my knowledge this remains a possible approach, should the authors elect to pursue it. This is certainly the approach that I would applaud. Suggestions for this are below. That there is an approved protocol is immaterial. The protocol can be amended to address what are presently material problems.

B. The authors can retain much of the current structure, but include a far more full and open acknowledgment of the suite of quite significant concerns and limitations in the present study design, and the limitations in the inferences that such a study can validly permit. This must be accompanied by amended analyses. Limitations and analyses that should be included are elaborated below.

If neither of these occur, in my view the manuscript should not be published. The result will be not merely a low quality product, but an approach that disrespects and may be construed to abuse patients’ time, effort, contribution and potential for suffering, via a design that can be expected to foster flawed inferences in a fashion that may place these and other patients at future risk. Whether or not an ethics board has approved the study, the study will not be ethical.

VI. Approaches to improve the study (option “a” above):

A. Account for potency. Potency dependence for statin adverse effects has been shown (as previously referenced) in multiple different study designs, not limited to but including head to head RCTs of more intensive vs less intensive statin therapy¹¹⁻¹³. (That Dale et al citation an abstract does not reduce its relevance; it is peer reviewed and includes data on CK elevation on statins in more vs less potent statin head to head RCT comparisons. The validity of its findings is buttressed by the full-publication Silva paper.)

Our own observational study is the only study to look at recurrence rates of muscle adverse effects of statins as a function of rechallenge after adverse effect with higher vs lower potency statins, looking at adverse effects initially occurring across statins and potencies¹⁴ – that is, it provides data in a setting most similar to that of participants that StatinWISE will presumably enroll. (a) participants had no basis to alter their representation of adverse effect occurrence based on relative potencies, and did not know that within-patient potency comparisons would be undertaken. b) The highest quality randomized evidence on the matter has found that blinding of patients does not materially alter patient reports: Meta-analyses from the Nordic Cochrane Center analyzing randomized trials that included a randomized blinded placebo arm (in which some patients may believe themselves to be on the active drug) and a randomized open-label no-treatment arm finds that there is little difference between the two¹⁵. Large “placebo effects”, then, arise not from patients’ response to beliefs about effects of a drug but from other factors – such as regression to the mean.

c) Though like every study that one has limitations, that study provides the highest quality extant evidence to directly examine within-patient recurrence of statin AEs as a function of potency, spanning a range of original statin drugs and potencies associated with adverse effects. It may also provide the best evidence related to timecourse of onset and recovery of statin muscle AEs. There is not higher quality evidence that repudiates or countermands it. But again, if the authors elect to continue to malign those data or characterize them as “opinion,” in any case potency relationships for statin AEs are also supported as above by double-blind head to head comparisons of more vs less intensive statins¹¹. e) Of note, even if there were no evidence of a potency relationship, a quality design would need to incorporate the possibility that a dose-relationship of adverse effects may be present. Dose relationships of drug and toxin adverse effects are a standard well-recognized consideration, and even if there had there been absence of evidence, this would in no way constitute evidence of absence.}

Addressing potency can be done in any of several ways.

1) A single drug-dose could be retained, but the sample restricted to those with past AEs on lower potency agents than will be used in this study.

2) Agents of multiple potencies could be included to ensure rechallenge potency was higher than potency of the agent on which the participant previously experienced adverse effects. There will still be the possibility of bias to the null from participants who had genetic factors or concurrent medications or temporally confined factors (like high grapefruit juice consumption for CYP3A4 metabolized statins) that affected their blood levels or response to the prior drug differentially relative to the present one. However our data suggest that use of higher potency agents would go far to reducing this.

3) Re-power the study with the assumption that the dose will be too low for some patients; be sure that the power adequately considers this. However this will require more assumptions. Options 1) or 2) are far superior.

B. Enhance prospects for reproducibility. A design that will have greater authority will be that which selectively enrolls participants who experienced resolution with dechallenge, and recurrence with rechallenge

(then, in order to participate, went off again with resolution) in the past. This will focus assessment on the group that will have the highest prospect of having had true statin AEs.

As a side benefit, the study will then also demonstrate whether past reproduction in an unblinded setting predicts future reproduction of symptoms in a blinded setting. (Here I assume that the potency issue has been properly addressed.) This would be a tremendous benefit, and radically reduce potential future costs and hassles for assessing the role of statins when apparent adverse effects arise. The cost to enroll each such participant would be higher with the focus placed on this group, but this would be offset by the fact that the number needed for adequate power, to show adverse effect reproduction, would be materially lower. Additionally costs would be saved by limiting the crossovers as below.

C. Ensure more adequate duration of each treatment period. The authors now cite the Cham 2010 paper in support of their two month time periods (though they spurn it for its evidence on recurrence with lower vs higher potency agents). They state “Two months is sufficient time for symptoms to appear in most patients and to washout from the previous treatment period (median time to symptom improvement was 2 weeks following cessation).

14" Two weeks was median time to first suggestion of improvement. Median time to maximal improvement after discontinuation, after the first rechallenge reproducing adverse effects, was 10 weeks (mean 27 weeks). This means more than half the patients did not experience maximum recovery within the two month time this study plans to allocate. (Of note, mean time to first evidence of symptom improvement was twice as long after rechallenge as after first statin causing problems.) This will provide serious potential for cross-over effects that will compromise the study. Allowing more adequate time can be done without extending the study participation period – indeed, contracting it – by reducing the treatment periods to one – or one crossover.

D. Limit the number of treatment phases, to one crossover, or alternatively one (parallel-design) phase, randomizing patients to statin or placebo in first phase cross-over studies are known to be materially compromised by carry-over effects¹⁶; and data from statins, including biopsy evidence, make clear that both clinical and physiological persistence (i.e. for such a design, carryover effects) may be present in those who experience problems on statins, with physiological persistence shown for both muscle and for nerve damage that may cause muscle symptoms¹⁷⁻¹⁹. A single phase or a single cross-over will reduce the risk that a state of intermediate injury is persistent and prevents discrimination between statin and placebo phases. "It also better protects participant safety. In one study that included "136 patients with drug-induced myopathies,...Control groups included 116 patients on therapy with no myopathic symptoms, 100 asymptomatic individuals from the general population never exposed to statins, and 106 patients with non-statin-induced myopathies" it was reported that "Variable persistent symptoms occurred in 68% of patients despite cessation of therapy"¹⁸. That study employs the term myopathy as muscle specialists do, to refer to presence of muscle injury and symptoms, without requirement for CK elevation – rather than via the revisionist definition employed in the statin RCT literature, that counts a problem as myopathy only if CK levels are 10x ULN.

E. Attend better to patient safety. The current approach considers only CK levels and ALT in determining the approach when patients develop symptoms. Yet previous studies have clearly shown that lack of CK elevation does not rule out structural muscle injury in those experiencing muscle symptoms on statins^{20, 21}.

E.g. in the double-blind cross-over biopsy study, N = 4, "Creatine kinase levels were normal in all four patients despite the presence of significant myopathy²¹". In another, N = 44, it was concluded that "A lack of elevated levels of circulating creatine phosphokinase does not rule out structural muscle injury"²⁰. In the above study showing variable persistent symptoms in 68% despite cessation of therapy, "In 52% of muscle biopsies from patients, significant biochemical abnormalities were found in mitochondrial or fatty acid metabolism, with 31% having multiple defects."¹⁸. Some clearly develop symptoms on statins because of such preexisting defects; however once triggered by statins, such symptoms can then persist, sometimes not following the first but following later usages. While these studies are not double-blinded, the information they provide is not "opinion" or "personal preference" and it is not supplanted by any more authoritative evidence. These studies make clear that real problems that can lead to persistent discomfort or disability can arise on statins in absence of CK elevation. Patients developing symptoms in the absence of CK elevation must also be treated as having, potentially, muscle injury. As per D, the best approach from the perspective of participant protection is to terminate participation

after one phase, or one cross-over – counting study discontinuation due to symptoms as an outcome.

F. Remove analysis barriers to seeing effects, if effects are present. In the sample size calculation please give the assumptions that underlie the simulation. “Using simulation, we estimated that a sample size of 64 participants provides approximately 90% power to detect a treatment effect of at least 1cm, assuming a Type I error of 5%.” I assume this analysis demands that the average effect across all participants and every cross-over is at least 1cm? But the basis of the 1cm is that this is meaningful for an individual patient, even if statins don't produce it in everyone. And the assumption of the study is that not everyone that has symptoms on statins has them due to statins. So requiring the average effect size to be at least 1cm makes an excessive demand if symptoms are reproduced only in a subset – which is relatively ensured by factors including failure to use an adequate statin potency in all participants.

An acceptable approach would be to count a signal as present or not within a patient at 1cm change, then assess the proportion that show that size change as increase, and the proportion that show that magnitude decrease, on statin relative to placebo (in a single phase or a single crossover); and complement this with a t-test of difference in mean change score on statin vs placebo – parallel-design first phase, and paired-test within participant in the one cross-over. In the present setting, in any case, the requirement that the average effect surpass a minimum effect size designated for clinical relevance to an individual is clearly inappropriate – since a basis for the study is that a true effect may not occur in all participants; and since design features like failure to consider potency ensure that. {Ordinarily power calculations are based on some form of preliminary evidence from the literature. What are the assumptions regarding expectation about fraction of patients affected and expected typical effect size? I am not discerning any contribution of evidence from the statin literature to this sample size calculation. }

G. I would urge inclusion of a randomized double blinded no-treatment group, that receives placebo in each cycle; with assessment of patient status both at conclusion of study participation and six and twelve months later.

This would allow assessment of whether rechallenging people repeatedly with statins, who may have previously had statin symptoms, may lead to worse outcomes. Participants should be asked if they feel their participation was helpful to them (which you do currently ask); they should be asked if they feel that participation was injurious to them. (Because of features of the sample that bias to the null, this will be a low-ball estimate, but still important to secure.) This would provide a crucial safety check of relevance to future N-of-1 trials. This assessment can seek to include participants who dropped from the clinical portion of the study.

VII. Approaches to retaining the basic design but improving the analyses and limitations. Ordinarily I am amenable to publishing many studies (since all studies have limitations) provided that the limitations are fully and openly acknowledged. I am concerned here by the stratagems employed in the design and the response, the failure to acknowledge some key issues, and the potential for peril to patients that may arise from this. However, the following would markedly improve the publishability of the study, retaining much of the present design. Alter the analysis approach.

Assess as above the fraction of individuals who meet the threshold – rather than requiring it as an overall effect, and examine the difference in mean change.

A. The authors state in their reply “Importantly, the main aim of StatinWISE is to determine whether muscle symptoms during statin use are caused by statins in particular individuals.” And “Study protocol for Statin Web-based Investigation of Side Effects (StatinWISE), a series of randomised controlled N of 1 trials comparing atorvastatin and placebo in UK primary care.” Based on these, the results of the individual N-of-1 trials need to be presented. You can share the fraction who meet the 1cm threshold, at the first cross-over (or first parallel-design comparison). Also assess t-test of mean parallel and crossover (paired-t) statin vs placebo difference. Prioritize assessment of the first phase (parallel-design) and first crossover. Based on significant risk of carryover effects compromising study outcomes, in my view the study should be restricted to that; but if it is not, those findings should be emphasized.

B. Incorporate prespecified subset analysis focused on participants with greatest expectation of reproduction: past history of dechallenge with improvement and rechallenge with recurrence, each within a shorter timeframe than you allot (ideally within 1 month), previously occurring on a lower potency statin agent. Ideally overrecruit this group to better power for them. Focus this analysis on the first time period. Compare results in that group to participants for whom none of the above are true. Then conduct exploratory analyses to examine the impact of each of the component considerations separately (past relative statin potency, past history of rechallenge with recurrence, short timeframe occurrence and resolution), and first vs later timeframes. Particularly if the study will not power adequately for this group, openly acknowledge that the study is underpowered to address the group with the strongest expectation of causal effects and that the study has inherent bias to the null.

C. For patients who drop from the study, do a careful job following-up reason for discontinuation. Be clear up front with patients that if they develop problems you wish to know about them if they discontinue (or if they don't). Because of the plan for crossover and the short treatment periods, it may not be possible to meaningfully interpret differences if any dropout on statin vs placebo: It would not be possible to predict, with two month periods, whether true AEs arising partway into statin use would lead to drops more often while still on statin or after transition to placebo (when symptoms have recently worsened with continued statin use and then not resolved promptly on crossover).

D. I would urge that patients also be given contact information for, and be urged to report any problems and dropouts to, an independent entity not associated with the study – such as an adverse effects reporting agency to which they should also mention their study participation. Based on these authors' approach to responding to comments, I have significant concerns in relation to how the information will be managed. (A documented consideration is that some studies, that purported to address drug adverse events, published favorable safety findings but were ascertained to have apparently manipulated, omitted or erased adverse events from the record, that had in reality supported problems²². I am hopeful but not confident that any such concern is unfounded in this case.

Urging patients to report directly and independently (without passing through study investigators) to an independent entity, in addition to the study (and providing the means for them to do so easily), would help to assuage such concerns.

E. Participants, including those who drop out, should be asked at conclusion of participation, and then again several months later, whether they believe participating in this study was helpful to them; and whether it was injurious to them. (Presumably it could be both.) While the information would be more helpful with inclusion of an blinded randomized placebo-only group, it will provide important information of relevance to rechallenge, even if such an arm is not incorporated.

F. Optional: Examine the relation between time since last statin dechallenge, prior to study participation, and time to (and occurrence of) symptom onset. Those with a longer time off statins might have a longer time, on average, to recurrence, if there has been greater recovery of underlying injury. Though, this study may not be sizeable enough to show this if statistically true, it should be assessed.

G. The Limitations have been improved. However further limitations should be acknowledged, as also indicated above.

1. It should be stated that persons who feel confident that their symptoms are due to statins may disproportionately elect not to participate, so the study may disproportionately consider more equivocal cases, conferring bias to the null.

2. It should be stated that patients may be placed on a lower potency statin than to which they previously cited problems, and that adverse effects are reported to be dose-dependent, conferring bias to the null.

3. It should be noted that the human body cannot be presumed to operate as a "meter" in assessing statin adverse effects – and that time for recovery may not be adequate for all. Such effects do not necessarily reliably turn off and on rapidly in response to withdrawal and rechallenge with a drug. We do not require of allergy or asthma sufferers that the patient is subjected to reproducibility testing of potential triggers in a controlled condition, or that they always respond the same way. (They do not.) Both for those conditions, and for statin adverse effects, the underlying physiology is subject to other influences that may alter true responses to a drug from one time to another.

4. It should be directly stated that given these biases, which operate primarily to the null, this study may be expected to understate the fraction of people who report adverse muscle effects on statins in whom the original cited problems were due to statins.

Specific statements:

There are many comments I could make to specific statements. I will restrict myself to a few.

I. "Importantly, the main aim of StatinWISE is to determine whether muscle symptoms during statin use are caused by statins in particular individuals."

A. That appears to be countermanded by the authors' statement – made twice in their response – that we don't know how often statins cause muscle problems and that is why this effort is being undertaken. That statement implies that there is an intent to draw inferences beyond these particular individuals.

B. How often statins cause such problems is inherently not a well-posed question, since effect modification and dose dependence mean there is not one risk level that applies to all.

It is still less well-posed with this study design, given expected self-selection bias, and biases to the null, that as above are in operation here.

(I appreciate that the authors now acknowledge that persons with more significant symptoms may not participate, however.)

Moreover, the present design does not permit inference to be drawn, either, about all of these particular individuals, except where adverse effects are reproduced. This is because the authors have elected not to address critical evidence as well as sound considerations (not personal preference) that influence whether real adverse effects would be reproduced in this study. Most seminally among these, adverse effects on statins are potency-dependent including (but not limited to) head to head more vs less intensive statin therapy in randomized trials^{11-13, 23}, but also shown in other study designs.

II. "We previously summarised high-quality randomised evidence regarding this relationship in our introduction"

Many of those data were not designed for this purpose, and are not high quality for that purpose. It is not methodologically sound to presume that randomized evidence is always high quality, for every purpose including that for which it was not designed.

A. Phenomena of selection and self-selection bias often strongly influence study participant characteristics²⁴⁻²⁶, in a direction that influences prospects of RCTs to show adverse effects – via the phenomenon of effect modification. Selection/ participation bias and effect modification are acknowledged methodological issues, not matters of personal preference. Study participants differ from nonparticipants – including but not limited to lower prevalence of other drugs and comorbidities, that are documented (not due to my "opinion") to increase risk of drug and specifically statin adverse effects (based on data – citations given previously (and a review here²⁷) – not opinion). This is not just via exclusion criteria but also self-selection.

B. It is not my "personal preference" that most statin RCTs are industry-funded, nor that both researchers and the FDA have repeatedly found – based on data, – that industry-funded trials can radically skew evidence on benefit and harm^{7, 8, 22, 28}. This is an issue about which former BMJ editor-in-chief Richard Smith was outspoken²⁹. As he said in reference to findings of a study, "It suggests that, far from conflict of interest being unimportant in the objective and pure world of science where method and the quality of data is everything, it is the main factor determining the result of studies."

Because of such issues, industry-funded RCTs that were not even designed to assess adverse effects, are for many reasons not high-quality studies for adverse effect ascertainment.

Far from being free from bias they bear potentially decisive forms of bias relative to this outcome type. (Based on evidence, not personal preference)

III. "However, the StatinWISE trial was scientifically peer reviewed prior to being funded by the UK NIHR and has been reviewed and approved by both the UK-MHRA and Ethics Committee."

This is also a scientific peer review, and by someone expressly knowledgeable about statin adverse effects. It is still possible to improve the protocol to help it more fully meet its stated purpose – to make it both more scientifically sound and more ethically acceptable.

	The mischaracterizations by the authors, of references that had been provided that support the existence of statin muscle effects seem to be aligned with an apparent reluctance to acknowledge evidence of relevance to the existence of adverse effects. When coupled with the analysis approach that – if I understand correctly – requires as an average effect that which is said to be relevant if present in any individual, as above, this raises questions regarding the intent of this study. “Ref 1. This is a study describing four case reports. The RCT from which they came is not cited in the article. We are unable to find any subsequent published RCT, so it is unclear what the umbrella RCT design was, or what its results were.” A. These findings are self-contained. The larger study is referenced as the source of participants who reported muscle problems on statins. The terminology of N-of-1 trials was not widely in use at that time: though Dr. Phillips refers to the four as case reports, the individuals participated in a set of assessments in blinded statin and placebo periods, analogous to your proposed study but with a number of critical advantages, including that participants not only reported symptoms but underwent (blinded) biopsy and muscle strength assessments, objective assessments that the StatinWISE study lacks. “Ref 5. We are aware that there are genetic variants associated with myopathy, but this is a more severe symptom than the ones we are investigating in StatinWISE and therefore have not cited this paper. We have clearly stated that statins are associated with myopathy on page 5 (Introduction).” A. Let me first comment on the issue of myopathy. It is defined one way by muscle and mitochondrial specialists, to refer to pathology or injury to muscle (myo-pathy) with or without CK elevation. It has been defined in a very different way in the statin RCT literature, to require CK elevations of at least 10x the upper limit of normal. But the former definition, the proposed StatinWISE study certainly does not exclude patients with myopathy. By the latter definition, the cited study assesses cases short of myopathy, of potential relevance to your study, not with “more severe symptom” but in some cases with less. The purpose of referencing this study was not primarily to indicate that there are genetic variants associated with myopathy. The purpose is several-fold. 1. The paper supports muscle problems (ones that do not meet criteria that these authors are presumably using for “myopathy”, i.e. 10xULN CK). This study includes assessment of (and evidence of a statin association for) 3xULN CK plus a hepatic finding, adding evidence for a muscle injury that does not meet criteria used for myopathy. That they counted this only when liver function criterion is also met does not undo the existence of the muscle effect. The fact that the authors employed the term “incipient myopathy” is immaterial to what is actually defined. This condition was assessed based on laboratory findings (including the 3xULN for CK) that do not meet the 10xULN criteria for myopathy, and it was deemed to be present “irrespective of whether there were muscle symptoms.” So what was assessed is not “a more severe symptom than the ones we are investigating”; some of the symptoms will be less severe. 2. The paper (though reflected in the abstract for the paper) adds data supporting a dose-relation for muscle problems, not restricted to the 10xULN definition of “myopathy”.
--	---

3. Elevated risk of statins AEs in persons with genetic variants relative to without (including of the non 10xULN “myopathy” type above) is a reminder that there is a risk of muscle problems due to statins to be elevated. Of note, other studies have shown genetic factors associated with risk of (biopsy documentable) muscle pathology on statins that also do not meet the industry-defined 10xULN criterion for “myopathy” (previously referenced 18, 20, 21). “We have now clarified our aim on page 6 and stated on page 4 the limitation of the study assessing a single statin at a single dose (which it is worth noting is the most commonly used statin at the most commonly used dose).” I appreciate this clarification, that the study addresses a single statin at a single dose, and acknowledge that it is a material improvement in the limitations. I observe, however, that the statement remains at odds with both the (repeated) characterization by the authors that they are doing the study because we do not know how often statins cause muscle problems (suggesting that the intent is to address this) “As stated above, the empirical scientific evidence to date shows that we do not know how often statins cause muscle symptoms: hence the need for this study.”; and also with the statement that “the main aim of StatinWISE is to determine whether muscle symptoms during statin use are caused by statins in particular individuals.”

Summary comments:

The paper could be made acceptable for publication, scientifically and ethically, only if the analysis approach is changed. This must include elimination of a demand for a 1cm average change, which (if I am interpreting that correctly, and I may not be) makes unacceptable demands on the subset in whom symptoms will recur on statin, if not all have symptoms reproduced. And it is a presumption of the study that not all necessarily will. This analysis might be revised to include assessment of the fraction who experience a positive 1cm change, and a negative 1cm change, on statin vs placebo, but this should be complemented by t-tests of mean difference in symptom rating on statin vs placebo. These assessments should be focused on the first phase (parallel-design) and first crossover (paired t-test) to limit the impact of carryover effects. The authors should also incorporate subset (first phase) analyses focused on those with strongest expectations of effect reproduction (prior rechallenge evidence at a lower statin potency), and should qualitatively compare results in this group to participants without rechallenge evidence whose symptoms were previously identified to arise only on a higher potency statin. Attention to dropouts (and cause for dropout) must be shored up, and there should be longer term follow-up of patient self-rated impact of study participation after participation discontinuation for all participants, for safety/ ethics reasons. And the limitations must be far more openly and fully acknowledged. It would be a far better, as a paper and a study, scientifically and ethically, if key elements of the design, beyond analysis, were also changed – particularly, the study should be altered to be limited to one parallel-phase or a single-crossover (the latter will typically have superior power); with participants allocated to a potency exceeding that which previously caused the problem (or the exact drug and dose that produced problems, as a second best and potentially safer – but with expectation that some “true” problems will not be reproduced), focused on those who previously experienced statin rechallenge with reproduction of symptoms, employing slightly longer study phases (e.g. three months – although this is less critical with a single cross-over),

-incorporating a randomized double-blinded placebo-only group to allow superior assessment of the impact of statin use on dropout, and then of longer term effects after undergoing rechallenging on statin (or not), in those who previously experienced muscle problems on statins.

References

1. Adams B. A group of influential doctors is calling on the government to tighten up NICE's conflicts of interests policy. *Pharmafile* 2014;Oct 22.
2. CBS Broadcasting Inc. New cholesterol advice tainted. www.cbsnews.com/stories/2004/07/12/health/main629058.shtml - 55k - Feb 14, 2005 2004;Download date 2-15-05.
3. Choudhry NK, Stelfox HT, Detsky AS. Relationships between authors of clinical practice guidelines and the pharmaceutical industry. *Jama* 2002;287:612-7.
4. Taylor P. A survey of UK doctors has found that many are refusing to apply NICE advice to offer statin therapy to low-risk patients with high cholesterol. *Pharmafile* 2014;March 11:<http://www.pharmafile.com/news/195552/uk-doctors-are-ignoring-statin-guidance>.
5. Taylor P. UK doctors are ignoring statin guidance. A survey of UK doctors has found that many are refusing to apply NICE advice to offer statin therapy to low-risk patients with high cholesterol. *Pharmafile.com* 2014.
6. Turner R, Etienne N, Alonso MG, et al. Antioxidant and anti-atherogenic activities of olive oil phenolics. *Int J Vitam Nutr Res* 2005;75:61-70.
7. Turner EH, Matthews AM, Linardatos E, Tell RA, Rosenthal R. Selective publication of antidepressant trials and its influence on apparent efficacy. *N Engl J Med* 2008;358:252-60.
8. Bero L, Oostvogel F, Bacchetti P, Lee K. Factors Associated with Findings of Published Trials of Drug-Drug Comparisons: Why Some Statins Appear More Efficacious than Others. *PLoS Med* 2007;4:e184.
9. Heres S, Davis J, Maino K, Jetzinger E, Kissling W, Leucht S. Why Olanzapine Beats Risperidone, Risperidone Beats Quetiapine, and Quetiapine Beats Olanzapine: An Exploratory Analysis of Head-to-Head Comparison Studies of Second-Generation Antipsychotics. *Am J Psychiatry* 2006;163:185-94.
10. Stelfox HT, Chua G, O'Rourke K, Detsky AS. Conflict of interest in the debate over calcium-channel antagonists. *N Engl J Med* 1998;338:101-6.
11. Silva M, Matthews ML, Jarvis C, et al. Meta-analysis of drug-induced adverse events associated with intensive-dose statin therapy. *Clin Ther* 2007;29:253-60.
12. Dale KM, Henyan NN, Coleman CI, Klueger J, White CM. Does more aggressive statin therapy increase muscle and liver risk? Abstract 981-193. *JACC* 2006;Feb 21:335 A.
13. Preiss D, Seshasai SR, Welsh P, et al. Risk of incident diabetes with intensive-dose compared with moderate-dose statin therapy: a meta-analysis. *JAMA* 2011;305:2556-64.
14. Cham S, Evans MA, Denenberg JO, Golomb BA. Statin-associated muscle-related adverse effects: a case series of 354 patients. *Pharmacotherapy* 2010;30:541-53.
15. Hrobjartsson A, Gotzsche PC. [What is the effect of placebo interventions? A systematic review of randomized clinical trials with placebo treated and untreated patients]. *Ugeskr Laeger* 2002;164:329-33.

	16. Mills EJ, Chan AW, Wu P, Vail A, Guyatt GH, Altman DG. Design, analysis, and presentation of crossover trials. Trials 2009;10:27. 17. Phillips PS, Haas RH. Observations from a statin myopathy clinic. Arch Intern Med 2006;166:1232-3. 18. Vladutiu GD, Simmons Z, Isackson PJ, et al. Genetic risk factors associated with lipid-lowering drug-induced myopathies. Muscle Nerve 2006;34:153-62. 19. Phan T, McLeod JG, Pollard JD, Peiris O, Rohan A, Halpern JP. Peripheral neuropathy associated with simvastatin. Journal of Neurology, Neurosurgery and Psychiatry 1995;58:625-8. 20. Mohaupt MG, Karas RH, Babiychuk EB, et al. Association between statin-associated myopathy and skeletal muscle damage. CMAJ 2009;181:E11-8. 21. Phillips PS, Haas RH, Bannykh S, et al. Statin-associated myopathy with normal creatine kinase levels. Ann Intern Med 2002;137:581-5. 22. Harris G. Caustic Government Report Deals Blow to Diabetes Drug. New York Times 2010;July 9, 2010. 23. Link E, Parish S, Armitage J, et al. SLCO1B1 variants and statin-induced myopathy—a genomewide study. N Engl J Med 2008;359:789-99. 24. Antman K, Amato D, Wood W, et al. Selection bias in clinical trials. J Clin Oncol 1985;3:1142-7. 25. Ganguli M, Lytle ME, Reynolds MD, Dodge HH. Random versus volunteer selection for a community-based study. J Gerontol A Biol Sci Med Sci 1998;53:M39-46. 26. Jennens RR, Giles GG, Fox RM. Increasing underrepresentation of elderly patients with advanced colorectal or non-small-cell lung cancer in chemotherapy trials. Intern Med J 2006;36:216-20. 27. Golomb BA, Evans MA. Statin adverse effects : a review of the literature and evidence for a mitochondrial mechanism. Am J Cardiovasc Drugs 2008;8:373-418. 28. Bero L. Industry sponsorship and research outcome: a Cochrane review. JAMA Intern Med 2013;173:580-1. 29. Smith R. Conflicts of interest: how money clouds objectivity. J R Soc Med 2006;99:292-7.
--	---

VERSION 2 – AUTHOR RESPONSE

Reviewer: 1

Reviewer Name: Gabriel Chodick

Institution and Country: Maccabi Healthcare Services, Israel

Competing Interests: No competing interests.

The authors have adequately addressed my comments.

Reviewer: 2

Reviewer Name: Johann Auer, MD

Institution and Country: Department of Cardiology, St. Josef Hospital Braunau, Austria

Competing Interests: none

Comment: This study focus on muscle symptoms. However, statins are associated with other skeletal-related symptoms beyond muscles. Assessment of such side effect would not increase burden on participants and GPs that much. In contrast, assessment of such symptoms would give further insights into this issue.

Response: We agree with the reviewer that statins have been linked to skeletal-related symptoms beyond muscles. However, we chose to minimize burden to patients by focusing on a single question relating to muscle symptoms. We are not able to address issues relating to the design of the study and cannot make changes to this.

Comment: Assessment of LDL-C is not an unusual procedure during treatment with a cholesterol-lowering drug. Most patients are aware of LDL-C at baseline and of changes of LDL levels during treatment. Thus, blinding may be compromised by (unscheduled) LDL-C. measurements during the study. This important issue remains and should be addressed.

Response: As we mentioned in our previous response to reviewers, we feel that repeated LDL measurements are unlikely in UK primary care. This is supported by current NICE guidelines, which recommend annual checks (NICE guideline CG181, Cardiovascular disease: risk assessment and reduction, including lipid modification). In practice, patients were found to have a mean of 1.3 cholesterol tests per year and over a third of patients had no tests in the year following statin initiation (Phatak, Value Health 2008 11(5):933). While we do not feel that frequency of testing will therefore impact blinding, this is certainly an issue that we will be able to raise in our Discussion.

Reviewer: 3

Reviewer Name: Beatrice A. Golomb, PhD

Institution and Country: University of California, San Diego, La Jolla, CA, United States

Competing Interests: I am the executor of an estate (and will be among the beneficiaries) that has some stock in statin manufacturing pharmaceutical companies. I have been advised by the estate lawyer to make no changes to the stock portfolio until the time of distribution.

Broad statements/ Responses:

I. Comment on reply opening: The authors begin their reply not by a point by point response to the substance of comments, but by a sweeping false and offensive characterization of the reviewer – who gave considerable time, and shared strong expertise on statin effects and study methodology, for no compensation, to provide information and suggestions of relevance to improving the authors' manuscript and the underlying study. While maligning the referee so as not to be responsible for fully addressing the content of the referee comments may be an effective rhetorical tool, the approach also constitutes an acknowledged logical fallacy – colloquially, “poisoning the well.” It is also highly unprofessional. The points I made in no instance reflect “personal preference” as the authors assert, except preference for high quality evidence and inference, and patient protection – each of which should be the authors' preferences as well. Nor do the comments reflect opinion other than that guided by sound inference from a more in depth knowledge of the evidence on this topic than most will have (including, almost certainly, those who approved the study proposal, as the latter are not selected for their expertise on this specific issue). Moreover in virtually all instances a literature foundation for statements was cited. (If there is any point which the reviewers feel to be lacking in cited evidence, I will be happy to share the undergirding evidence if asked.) The authors comments about evidence suggest a belief that existing RCTs are free from bias and inherently and always provide high quality evidence to address every question, moreso than other study designs – and irrespective of what those RCTs were designed to address, how they were designed, what selection characteristics they had, what outcomes they assessed and how, and who the funders were.

This is a simplistic and fallacious perspective, that fails to grasp the impact of key factors such as self-selection and selection bias that can seriously compromise external validity; and the well-documented and often profound impact of drug company conflict of interest on reported study results and conclusions. These are not issues of my “personal preference”; these are basic considerations of scientific methodology powerfully buttressed by empirical evidence. Apropos of this, if my inferences depart from those of others, this cannot be presumed to reflect unfavorably on me. The statin literature (and guidelines, as journalists have noted¹⁻³) is driven heavily by parties and studies with industry conflict of interest^{4, 5}. Data (not “personal preference” or “opinion”) show that such industry conflicts of interest are tied to not small, but often radical shifts in stated results⁶⁻⁸, and even moreso conclusions⁶⁻⁹ (as well as stated opinions¹⁰). {Of note, the groundbreaking experiments of pioneering social psychologist Solomon Asch show that once there are individuals who have willfully distorted results and conclusions, many others will believe the distorted conclusions even when this requires disregarding the obvious evidence before their eyes.}

II. I am sure it is frustrating to receive feedback that asks for additional effort for a study that has already been approved.

III. Please understand that my comments are intended to protect against study flaws including and especially those that are expected to bias results to the null and may lead to flawed inferences that may place future patients in needless peril. Better the authors’ frustration now, then harm to patients later. For this study more than for most, the importance of sound design cannot be overstated. A) The study asks that participants place themselves at possible risk, and this should only be done with the highest quality design, and the greatest protection against type II error. B) If the inferences drawn from the study are in error, in the direction of fostering false dismissal of patients’ adverse effect experiences, the study may lead to subjecting many future patients (potentially including these specific ones) to needless suffering. Because of this it is vital that the authors take measures to ensure the design does not foster false negative results. This study has potential to make an important contribution; I would like for it to do so.

IV. There are high quality data supporting all of the key points I had made – such as for dose-dependence of statin adverse effects, (self)-selection bias, effect modification, and the influence of conflict of interest on evidence. Even had these not been separately characterized in the case of statin adverse effects, they are known considerations in relation to study design and interpretation generally. That is, addressing limitations based upon these is not an expression of “personal preference” but appropriate recognition of important methodological considerations. (Understandably, the authors’ “personal preference” may be to discount such evidence selectively where it is disadvantageous.)

V. As I see it, the options are: A. The authors can put in effort now, to make this a better quality study – even a good quality study with improved prospects to address its supposed goals. To my knowledge this remains a possible approach, should the authors elect to pursue it. This is certainly the approach that I would applaud. Suggestions for this are below. That there is an approved protocol is immaterial. The protocol can be amended to address what are presently material problems. B. The authors can retain much of the current structure, but include a far more full and open acknowledgment of the suite of quite significant concerns and limitations in the present study design, and the limitations in the inferences that such a study can validly permit. This must be accompanied by amended analyses. Limitations and analyses that should be included are elaborated below. If neither of these occur, in my view the manuscript should not be published.

The result will be not merely a low quality product, but an approach that disrespects and may be construed to abuse patients' time, effort, contribution and potential for suffering, via a design that can be expected to foster flawed inferences in a fashion that may place these and other patients at future risk. Whether or not an ethics board has approved the study, the study will not be ethical.

VI. Approaches to improve the study (option "a" above):

A. Account for potency. Potency dependence for statin adverse effects has been shown (as previously referenced) in multiple different study designs, not limited to but including head to head RCTs of more intensive vs less intensive statin therapy¹¹⁻¹³. (That Dale et al citation an abstract does not reduce its relevance; it is peer reviewed and includes data on CK elevation on statins in more vs less potent statin head to head RCT comparisons. The validity of its findings is buttressed by the full-publication Silva paper.) Our own observational study is the only study to look at recurrence rates of muscle adverse effects of statins as a function of rechallenge after adverse effect with higher vs lower potency statins, looking at adverse effects initially occurring across statins and potencies¹⁴ – that is, it provides data in a setting most similar to that of participants that StatinWISE will presumably enroll.

{a) participants had no basis to alter their representation of adverse effect occurrence based on relative potencies, and did not know that within-patient potency comparisons would be undertaken. b) The highest quality randomized evidence on the matter has found that blinding of patients does not materially alter patient reports: Meta-analyses from the Nordic Cochrane Center analyzing randomized trials that included a randomized blinded placebo arm (in which some patients may believe themselves to be on the active drug) and a randomized open-label no-treatment arm finds that there is little difference between the two¹⁵. Large "placebo effects", then, arise not from patients' response to beliefs about effects of a drug but from other factors – such as regression to the mean. c) Though like every study that one has limitations, that study provides the highest quality extant evidence to directly examine within-patient recurrence of statin AEs as a function of potency, spanning a range of original statin drugs and potencies associated with adverse effects. It may also provide the best evidence related to timecourse of onset and recovery of statin muscle AEs. There is not higher quality evidence that repudiates or countermands it. But again, if the authors elect to continue to malign those data or characterize them as "opinion," in any case potency relationships for statin AEs are also supported as above by double-blind head to head comparisons of more vs less intensive statins¹¹. e) Of note, even if there were no evidence of a potency relationship, a quality design would need to incorporate the possibility that a dose-relation of adverse effects may be present. Dose relationships of drug and toxin adverse effects are a standard well- recognized consideration, and even if there had there been absence of evidence, this would in no way constitute evidence of absence.}

Addressing potency can be done in any of several ways. 1) A single drug-dose could be retained, but the sample restricted to those with past AEs on lower potency agents than will be used in this study. 2) Agents of multiple potencies could be included to ensure rechallenge potency was higher than potency of the agent on which the participant previously experienced adverse effects. There will still be the possibility of bias to the null from participants who had genetic factors or concurrent medications or temporally confined factors (like high grapefruit juice consumption for CYP3A4 metabolized statins) that affected their blood levels or response to the prior drug differentially relative to the present one. However our data suggest that use of higher potency agents would go far to reducing this.

3) Re-power the study with the assumption that the dose will be too low for some patients; be sure that the power adequately considers this. However this will require more assumptions. Options 1) or 2) are far superior.

B. Enhance prospects for reproducibility. A design that will have greater authority will be that which selectively enrolls participants who experienced resolution with dechallenge, and recurrence with rechallenge (then, in order to participate, went off again with resolution) in the past. This will focus assessment on the group that will have the highest prospect of having had true statin AEs.

As a side benefit, the study will then also demonstrate whether past reproduction in an unblinded setting predicts future reproduction of symptoms in a blinded setting. (Here I assume that the potency issue has been properly addressed.) This would be a tremendous benefit, and radically reduce potential future costs and hassles for assessing the role of statins when apparent adverse effects arise. The cost to enroll each such participant would be higher with the focus placed on this group, but this would be offset by the fact that the number needed for adequate power, to show adverse effect reproduction, would be materially lower. Additionally costs would be saved by limiting the crossovers as below.

C. Ensure more adequate duration of each treatment period. The authors now cite the Cham 2010 paper in support of their two month time periods (though they spurn it for its evidence on recurrence with lower vs higher potency agents). They state “Two months is sufficient time for symptoms to appear in most patients and to washout from the previous treatment period (median time to symptom improvement was 2 weeks following cessation).¹⁴” Two weeks was median time to first suggestion of improvement. Median time to maximal improvement after discontinuation, after the first rechallenge reproducing adverse effects, was 10 weeks (mean 27 weeks). This means more than half the patients did not experience maximum recovery within the two month time this study plans to allocate. (Of note, mean time to first evidence of symptom improvement was twice as long after rechallenge as after first statin causing problems.) This will provide serious potential for cross-over effects that will compromise the study. Allowing more adequate time can be done without extending the study participation period – indeed, contracting it – by reducing the treatment periods to one – or one crossover.

D. Limit the number of treatment phases, to one crossover, or alternatively one (parallel-design) phase, randomizing patients to statin or placebo in first phase cross- over studies are known to be materially compromised by carry-over effects¹⁶; and data from statins, including biopsy evidence, make clear that both clinical and physiological persistence (i.e. for such a design, carryover effects) may be present in those who experience problems on statins, with physiological persistence shown for both muscle

and for nerve damage that may cause muscle symptoms¹⁷⁻¹⁹. A single phase or a single cross-over will reduce the risk that a state of intermediate injury is persistent and prevents discrimination between statin and placebo phases. “It also better protects participant safety. In one study that included “136 patients with drug-induced myopathies,...Control groups included 116 patients on therapy with no myopathic symptoms, 100 asymptomatic individuals from the general population never exposed to statins, and 106 patients with non-statin-induced myopathies” it was reported that “Variable persistent symptoms occurred in 68% of patients despite cessation of therapy”¹⁸. That study employs the term myopathy as muscle specialists do, to refer to presence of muscle injury and symptoms, without requirement for CK elevation – rather than via the revisionist definition employed in the statin RCT literature, that counts a problem as myopathy only if CK levels are 10x ULN.

E. Attend better to patient safety. The current approach considers only CK levels and ALT in determining the approach when patients develop symptoms. Yet previous studies have clearly shown that lack of CK elevation does not rule out structural muscle injury in those experiencing muscle symptoms on statins^{20, 21}. E.g. in the double-blind cross-over biopsy study, N = 4, “Creatine kinase levels were normal in all four patients despite the presence of significant myopathy²¹”.

In another, N = 44, it was concluded that “A lack of elevated levels of circulating creatine phosphokinase does not rule out structural muscle injury”²⁰. In the above study showing variable persistent symptoms in 68% despite cessation of therapy, “In 52% of muscle biopsies from patients, significant biochemical abnormalities were found in mitochondrial or fatty acid metabolism, with 31% having multiple defects.”¹⁸. Some clearly develop symptoms on statins because of such preexisting defects; however once triggered by statins, such symptoms can then persist, sometimes not following the first but following later usages. While these studies are not double-blinded, the information they provide is not “opinion” or “personal preference” and it is not supplanted by any more authoritative evidence. These studies make clear that real problems that can lead to persistent discomfort or disability can arise on statins in absence of CK elevation.

Patients developing symptoms in the absence of CK elevation must also be treated as having, potentially, muscle injury. As per D, the best approach from the perspective of participant protection is to terminate participation after one phase, or one cross-over – counting study discontinuation due to symptoms as an outcome.

F. Remove analysis barriers to seeing effects, if effects are present. In the sample size calculation please give the assumptions that underlie the simulation. “Using simulation, we estimated that a sample size of 64 participants provides approximately 90% power to detect a treatment effect of at least 1cm, assuming a Type I error of 5%.” I assume this analysis demands that the average effect across all participants and every cross-over is at least 1cm? But the basis of the 1cm is that this is meaningful for an individual patient, even if statins don’t produce it in everyone. And the assumption of the study is that not everyone that has symptoms on statins has them due to statins. So requiring the average effect size to be at least 1cm makes an excessive demand if symptoms are reproduced only in a subset – which is relatively ensured by factors including failure to use an adequate statin potency in all participants.

An acceptable approach would be to count a signal as present or not within a patient at 1cm change, then assess the proportion that show that size change as increase, and the proportion that show that magnitude decrease, on statin relative to placebo (in a single phase or a single crossover); and complement this with a t-test of difference in mean change score on statin vs placebo – parallel-design first phase, and paired-test within participant in the one cross-over.

In the present setting, in any case, the requirement that the average effect surpass a minimum effect size designated for clinical relevance to an individual is clearly inappropriate – since a basis for the study is that a true effect may not occur in all participants; and since design features like failure to consider potency ensure that.

{Ordinarily power calculations are based on some form of preliminary evidence from the literature. What are the assumptions regarding expectation about fraction of patients affected and expected typical effect size? I am not discerning any contribution of evidence from the statin literature to this sample size calculation. }

G. I would urge inclusion of a randomized double blinded no-treatment group, that receives placebo in each cycle; with assessment of patient status both at conclusion of study participation and six and twelve months later. This would allow assessment of whether rechallenging people repeatedly with statins, who may have previously had statin symptoms, may lead to worse outcomes. Participants should be asked if they feel their participation was helpful to them (which you do currently ask); they should be asked if they feel that participation was injurious to them. (Because of features of the sample that bias to the null, this will be a low-ball estimate, but still important to secure.) This would provide a crucial safety check of relevance to future N-of-1 trials. This assessment can seek to include participants who dropped from the clinical portion of the study.

VII. Approaches to retaining the basic design but improving the analyses and limitations. Ordinarily I am amenable to publishing many studies (since all studies have limitations) provided that the limitations are fully and openly acknowledged. I am concerned here by the stratagems employed in the design and the response, the failure to acknowledge some key issues, and the potential for peril to patients that may arise from this. However, the following would markedly improve the publishability of the study, retaining much of the present design.

Alter the analysis approach. Assess as above the fraction of individuals who meet the threshold – rather than requiring it as an overall effect, and examine the difference in mean change.

A. The authors state in their reply “Importantly, the main aim of StatinWISE is to determine whether muscle symptoms during statin use are caused by statins in particular individuals.” And “Study protocol for Statin Web-based Investigation of Side Effects (StatinWISE), a series of randomised controlled N of 1 trials comparing atorvastatin and placebo in UK primary care.” Based on these, the results of the individual N-of-1 trials need to be presented.

You can share the fraction who meet the 1cm threshold, at the first cross-over (or first parallel-design comparison). Also assess t-test of mean parallel and crossover (paired-t) statin vs placebo difference. Prioritize assessment of the first phase (parallel-design) and first crossover. Based on significant risk of carryover effects compromising study outcomes, in my view the study should be restricted to that; but if it is not, those findings should be emphasized.

B. Incorporate prespecified subset analysis focused on participants with greatest expectation of reproduction: past history of dechallenge with improvement and rechallenge with recurrence, each within a shorter timeframe than you allot (ideally within 1 month), previously occurring on a lower potency statin agent. Ideally overrecruit this group to better power for them. Focus this analysis on the first time period. Compare results in that group to participants for whom none of the above are true. Then conduct exploratory analyses to examine the impact of each of the component considerations separately (past relative statin potency, past history of rechallenge with recurrence, short timeframe occurrence and resolution), and first vs later timeframes. Particularly if the study will not power adequately for this group, openly acknowledge that the study is underpowered to address the group with the strongest expectation of causal effects and that the study has inherent bias to the null.

C. For patients who drop from the study, do a careful job following-up reason for discontinuation. Be clear up front with patients that if they develop problems you wish to know about them if they discontinue (or if they don't). Because of the plan for crossover and the short treatment periods, it may not be possible to meaningfully interpret differences if any dropout on statin vs placebo: It would not be possible to predict, with two month periods, whether true AEs arising partway into statin use would lead to drops more often while still on statin or after transition to placebo (when symptoms have recently worsened with continued statin use and then not resolved promptly on crossover).

D. I would urge that patients also be given contact information for, and be urged to report any problems and dropouts to, an independent entity not associated with the study – such as an adverse effects reporting agency to which they should also mention their study participation. Based on these authors' approach to responding to comments, I have significant concerns in relation to how the information will be managed. (A documented consideration is that some studies, that purported to address drug adverse events, published favorable safety findings but were ascertained to have apparently manipulated, omitted or erased adverse events from the record, that had in reality supported problems²². I am hopeful but not confident that any such concern is unfounded in this case. Urging patients to report directly and independently (without passing through study investigators) to an independent entity, in addition to the study (and providing the means for them to do so easily), would help to assuage such concerns.

E. Participants, including those who drop out, should be asked at conclusion of participation, and then again several months later, whether they believe participating in this study was helpful to them; and whether it was injurious to them. (Presumably it could be both.) While the information would be more helpful with inclusion of an blinded randomized placebo-only group, it will provide important information of relevance to rechallenge, even if such an arm is not incorporated.

F. Optional: Examine the relation between time since last statin dechallenge, prior to study participation, and time to (and occurrence of) symptom onset. Those with a longer time off statins might have a longer time, on average, to recurrence, if there has been greater recovery of underlying injury. Though, this study may not be sizeable enough to show this if statistically true, it should be assessed.

G. The Limitations have been improved. However further limitations should be acknowledged, as also indicated above.

1. It should be stated that persons who feel confident that their symptoms are due to statins may disproportionately elect not to participate, so the study may disproportionately consider more equivocal cases, conferring bias to the null.

2. It should be stated that patients may be placed on a lower potency statin than to which they previously cited problems, and that adverse effects are reported to be dose-dependent, conferring bias to the null.

3. It should be noted that the human body cannot be presumed to operate as a “meter” in assessing statin adverse effects – and that time for recovery may not be adequate for all. Such effects do not necessarily reliably turn off and on rapidly in response to withdrawal and rechallenge with a drug. We do not require of allergy or asthma sufferers that the patient is subjected to reproducibility testing of potential triggers in a controlled condition, or that they always respond the same way. (They do not.) Both for those conditions, and for statin adverse effects, the underlying physiology is subject to other influences that may alter true responses to a drug from one time to another. 4. It should be directly stated that given these biases, which operate primarily to the null, this study may be expected to understate the fraction of people who report adverse muscle effects on statins in whom the original cited problems were due to statins.

Specific statements: There are many comments I could make to specific statements. I will restrict myself to a few.

Comment I. “Importantly, the main aim of StatinWISE is to determine whether muscle symptoms during statin use are caused by statins in particular individuals.”

Response: That appears to be countermanded by the authors’ statement – made twice in their response – that we don’t know how often statins cause muscle problems and that is why this effort is being undertaken. That statement implies that there is an intent to draw inferences beyond these particular individuals.

How often statins cause such problems is inherently not a well-posed question, since effect modification and dose dependence mean there is not one risk level that applies to all. It is still less well-posed with this study design, given expected self-selection bias, and biases to the null, that as above are in operation here. (I appreciate that the authors now acknowledge that persons with more significant symptoms may not participate, however.)

Moreover, the present design does not permit inference to be drawn, either, about all of these particular individuals, except where adverse effects are reproduced. This is because the authors have elected not to address critical evidence as well as sound considerations (not personal preference) that influence whether real adverse effects would be reproduced in this study. Most seminally among these, adverse effects on statins are potency-dependent including (but not limited to) head to head more vs less intensive statin therapy in randomized trials^{11-13, 23}, but also shown in other study designs.

Comment II. “We previously summarised high-quality randomised evidence regarding this relationship in our introduction”

Many of those data were not designed for this purpose, and are not high quality for that purpose. It is not methodologically sound to presume that randomized evidence is always high quality, for every purpose including that for which it was not designed.

Response: Phenomena of selection and self-selection bias often strongly influence study participant characteristics²⁴⁻²⁶, in a direction that influences prospects of RCTs to show adverse effects – via the phenomenon of effect modification. Selection/ participation bias and effect modification are acknowledged methodological issues, not matters of personal preference.

Study participants differ from nonparticipants – including but not limited to lower prevalence of other drugs and comorbidities, that are documented (not due to my “opinion”) to increase risk of drug and specifically statin adverse effects (based on data – citations given previously (and a review here²⁷) – not opinion). This is not just via exclusion criteria but also self-selection.

B. It is not my “personal preference” that most statin RCTs are industry-funded, nor that both researchers and the FDA have repeatedly found – based on data, – that industry- funded trials can radically skew evidence on benefit and harm^{7, 8, 22, 28}. This is an issue about which former BMJ editor-in-chief Richard Smith was outspoken²⁹. As he said in reference to findings of a study, “It suggests that, far from conflict of interest being unimportant in the objective and pure world of science where method and the quality of data is everything, it is the main factor determining the result of studies.”

Because of such issues, industry-funded RCTs that were not even designed to assess adverse effects, are for many reasons not high-quality studies for adverse effect ascertainment. Far from being free from bias they bear potentially decisive forms of bias relative to this outcome type. (Based on evidence, not personal preference)

Comment III. “However, the StatinWISE trial was scientifically peer reviewed prior to being funded by the UK NIHR and has been reviewed and approved by both the UK-MHRA and Ethics Committee.”

This is also a scientific peer review, and by someone expressly knowledgeable about statin adverse effects. It is still possible to improve the protocol to help it more fully meet its stated purpose – to make it both more scientifically sound and more ethically acceptable.

The mischaracterizations by the authors, of references that had been provided that support the existence of statin muscle effects seem to be aligned with an apparent reluctance to acknowledge evidence of relevance to the existence of adverse effects. When coupled with the analysis approach that – if I understand correctly – requires as an average effect that which is said to be relevant if present in any individual, as above, this raises questions regarding the intent of this study.

Comment: “Ref 1. This is a study describing four case reports. The RCT from which they came is not cited in the article. We are unable to find any subsequent published RCT, so it is unclear what the umbrella RCT design was, or what its results were.”

Response: These findings are self-contained. The larger study is referenced as the source of participants who reported muscle problems on statins. The terminology of N-of-1 trials was not widely in use at that time: though Dr. Phillips refers to the four as case reports, the individuals participated in a set of assessments in blinded statin and placebo periods, analogous to your proposed study but with a number of critical advantages, including that participants not only reported symptoms but underwent (blinded) biopsy and muscle strength assessments, objective assessments that the StatinWISE study lacks.

Comment: “Ref 5. We are aware that there are genetic variants associated with myopathy, but this is a more severe symptom than the ones we are investigating in StatinWISE and therefore have not cited this paper. We have clearly stated that statins are associated with myopathy on page 5 (Introduction).”

Response: Let me first comment on the issue of myopathy. It is defined one way by muscle and mitochondrial specialists, to refer to pathology or injury to muscle (myo-pathy) with or without CK elevation. It has been defined in a very different way in the statin RCT literature, to require CK elevations of at least 10x the upper limit of normal. But the former definition, the proposed StatinWISE study certainly does not exclude patients with myopathy. By the latter definition, the cited study assesses cases short of myopathy, of potential relevance to your study, not with “more severe symptom” but in some cases with less. The purpose of referencing this study was not primarily to indicate that there are genetic variants associated with myopathy. The purpose is several-fold.

1. The paper supports muscle problems (ones that do not meet criteria that these authors are presumably using for “myopathy”, i.e. 10xULN CK). This study includes assessment of (and evidence of a statin association for) 3xULN CK plus a hepatic finding, adding evidence for a muscle injury that does not meet criteria used for myopathy.

That they counted this only when liver function criterion is also met does not undo the existence of the muscle effect. The fact that the authors employed the term “incipient myopathy” is immaterial to what is actually defined.

This condition was assessed based on laboratory findings (including the 3xULN for CK) that do not meet the 10xULN criteria for myopathy, and it was deemed to be present “irrespective of whether there were muscle symptoms.” So what was assessed is not “a more severe symptom than the ones we are investigating”; some of the symptoms will be less severe.

2. The paper (though reflected in the abstract for the paper) adds data supporting a dose-relation for muscle problems, not restricted to the 10xULN definition of “myopathy”.

3. Elevated risk of statins AEs in persons with genetic variants relative to without (including of the non 10xULN “myopathy” type above) is a reminder that there is a risk of muscle problems due to statins to be elevated.

Of note, other studies have shown genetic factors associated with risk of (biopsy documentable) muscle pathology on statins that also do not meet the industry- defined 10xULN criterion for “myopathy” (previously referenced 18, 20, 21).

“We have now clarified our aim on page 6 and stated on page 4 the limitation of the study assessing a single statin at a single dose (which it is worth noting is the most commonly used statin at the most commonly used dose).”

I appreciate this clarification, that the study addresses a single statin at a single dose, and acknowledge that it is a material improvement in the limitations. I observe, however, that the statement remains at odds with both the (repeated) characterization by the authors that they are doing the study because we do not know how often statins cause muscle problems (suggesting that the intent is to address this) “As stated above, the empirical scientific evidence to date shows that we do not know how often statins cause muscle symptoms: hence the need for this study.”; and also with the statement that “the main aim of StatinWISE is to determine whether muscle symptoms during statin use are caused by statins in particular individuals.”

Summary comments: The paper could be made acceptable for publication, scientifically and ethically, only if the analysis approach is changed. This must include elimination of a demand for a 1cm average change, which (if I am interpreting that correctly, and I may not be) makes unacceptable demands on the subset in whom symptoms will recur on statin, if not all have symptoms reproduced. And it is a presumption of the study that not all necessarily will. This analysis might be revised to include assessment of the fraction who experience a positive 1cm change, and a negative 1cm change, on statin vs placebo, but this should be complemented by t-tests of mean difference in symptom rating on statin vs placebo. These assessments should be focused on the first phase (parallel-design) and first crossover (paired t-test) to limit the impact of carryover effects. The authors should also incorporate subset (first phase) analyses focused on those with strongest expectations of effect reproduction (prior rechallenge evidence at a lower statin potency), and should qualitatively compare results in this group to participants without rechallenge evidence whose symptoms were previously identified to arise only on a higher potency statin. Attention to dropouts (and cause for dropout) must be shored up, and there should be longer term follow-up of patient self-rated impact of study participation after participation discontinuation for all participants, for safety/ ethics reasons. And the limitations must be far more openly and fully acknowledged.

It would be a far better, as a paper and a study, scientifically and ethically, if key elements of the design, beyond analysis, were also changed – particularly, the study should be altered to be limited to one parallel-phase or a single-crossover (the latter will typically have superior power); with participants allocated to a potency exceeding that which previously caused the problem (or the exact drug and dose that produced problems, as a second best and potentially safer – but with expectation that some “true” problems will not be reproduced), focused on those who previously experienced statin rechallenge with reproduction of symptoms, employing slightly longer study phases (e.g. three months – although this is less critical with a single cross-over), incorporating a randomized double-blinded placebo-only group to allow superior assessment of the impact of statin use on dropout, and then of longer term effects after undergoing rechallenging on statin (or not), in those who previously experienced muscle problems on statins.

References

1. Adams B. A group of influential doctors is calling on the government to tighten up NICE's conflicts of interests policy. *Pharmafile* 2014;Oct 22.
2. CBS Broadcasting Inc. New cholesterol advice tainted. www.cbsnews.com/stories/2004/07/12/health/main629058.shtml - 55k - Feb 14, 2005 2004;Download date 2-15-05.
3. Choudhry NK, Stelfox HT, Detsky AS. Relationships between authors of clinical practice guidelines and the pharmaceutical industry. *Jama* 2002;287:612-7.
4. Taylor P. A survey of UK doctors has found that many are refusing to apply NICE advice to offer statin therapy to low-risk patients with high cholesterol. *Pharmafile* 2014;March 11:<http://www.pharmafile.com/news/195552/uk-doctors-are-ignoring-statin-guidance>.
5. Taylor P. UK doctors are ignoring statin guidance. A survey of UK doctors has found that many are refusing to apply NICE advice to offer statin therapy to low-risk patients with high cholesterol. *Pharmafile.com* 2014.
6. Turner R, Etienne N, Alonso MG, et al. Antioxidant and anti-atherogenic activities of olive oil phenolics. *Int J Vitam Nutr Res* 2005;75:61-70.
7. Turner EH, Matthews AM, Linardatos E, Tell RA, Rosenthal R. Selective publication of antidepressant trials and its influence on apparent efficacy. *N Engl J Med* 2008;358:252-60.
8. Bero L, Oostvogel F, Bacchetti P, Lee K. Factors Associated with Findings of Published Trials of Drug-Drug Comparisons: Why Some Statins Appear More Efficacious than Others. *PLoS Med* 2007;4:e184.
9. Heres S, Davis J, Maino K, Jetzinger E, Kissling W, Leucht S. Why Olanzapine Beats Risperidone, Risperidone Beats Quetiapine, and Quetiapine Beats Olanzapine: An Exploratory Analysis of Head-to-Head Comparison Studies of Second-Generation Antipsychotics. *Am J Psychiatry* 2006;163:185-94.
10. Stelfox HT, Chua G, O'Rourke K, Detsky AS. Conflict of interest in the debate over calcium-channel antagonists. *N Engl J Med* 1998;338:101-6.
11. Silva M, Matthews ML, Jarvis C, et al. Meta-analysis of drug-induced adverse events associated with intensive-dose statin therapy. *Clin Ther* 2007;29:253-60.
12. Dale KM, Henyan NN, Coleman CI, Klueger J, White CM. Does more aggressive statin therapy increase muscle and liver risk? Abstract 981-193. *JACC* 2006;Feb 21:335 A.
13. Preiss D, Seshasai SR, Welsh P, et al. Risk of incident diabetes with intensive-dose compared with moderate-dose statin therapy: a meta-analysis. *JAMA* 2011;305:2556-64.
14. Cham S, Evans MA, Denenberg JO, Golomb BA. Statin-associated muscle-related adverse effects: a case series of 354 patients. *Pharmacotherapy* 2010;30:541-53.
15. Hrobjartsson A, Gotzsche PC. [What is the effect of placebo interventions? A systematic review of randomized clinical trials with placebo treated and untreated patients]. *Ugeskr Laeger* 2002;164:329-33.
16. Mills EJ, Chan AW, Wu P, Vail A, Guyatt GH, Altman DG. Design, analysis, and presentation of crossover trials. *Trials* 2009;10:27.
17. Phillips PS, Haas RH. Observations from a statin myopathy clinic. *Arch Intern Med* 2006;166:1232-3.
18. Vladutiu GD, Simmons Z, Isackson PJ, et al. Genetic risk factors associated with lipid-lowering drug-induced myopathies. *Muscle Nerve* 2006;34:153-62.
19. Phan T, McLeod JG, Pollard JD, Peiris O, Rohan A, Halpern JP. Peripheral neuropathy associated with simvastatin. *Journal of Neurology, Neurosurgery and Psychiatry* 1995;58:625-8.

20. Mohaupt MG, Karas RH, Babiychuk EB, et al. Association between statin-associated myopathy and skeletal muscle damage. *CMAJ* 2009;181:E11-8. 21. Phillips PS, Haas RH, Bannykh S, et al. Statin-associated myopathy with normal creatine kinase levels. *Ann Intern Med* 2002;137:581-5. 22. Harris G. Caustic Government Report Deals Blow to Diabetes Drug. *New York Times* 2010;July 9, 2010. 23. Link E, Parish S, Armitage J, et al. SLCO1B1 variants and statin-induced myopathy—a genomewide study. *N Engl J Med* 2008;359:789-99. 24. Antman K, Amato D, Wood W, et al. Selection bias in clinical trials. *J Clin Oncol* 1985;3:1142-7. 25. Ganguli M, Lytle ME, Reynolds MD, Dodge HH. Random versus volunteer selection for a community-based study. *J Gerontol A Biol Sci Med Sci* 1998;53:M39-46. 26. Jennens RR, Giles GG, Fox RM. Increasing underrepresentation of elderly patients with advanced colorectal or non-small-cell lung cancer in chemotherapy trials. *Intern Med J* 2006;36:216-20. 27. Golomb BA, Evans MA. Statin adverse effects : a review of the literature and evidence for a mitochondrial mechanism. *Am J Cardiovasc Drugs* 2008;8:373-418. 28. Bero L. Industry sponsorship and research outcome: a Cochrane review. *JAMA Intern Med* 2013;173:580-1. 29. Smith R. Conflicts of interest: how money clouds objectivity. *J R Soc Med* 2006;99:292- 7.